# P53-dependent hypusination of eIF5A affects mitochondrial translation and senescence immune surveillance

Xiangli Jiang [1,2,10], Ali Hyder Baig[1,10], Giuliana Palazzo[1,3], Rossella Del Pizzo[1,3], Toman Bortecen[3,4], Sven Groessl [3,5], Esther A. Zaal [6], Cinthia Claudia Amaya Ramirez [1], Alexander Kowar [1,3], Daniela Aviles-Huerta[1,3], Celia R. Berkers [6], Wilhelm Palm [5], Darjus Tschaharganeh[7,8], Jeroen Krijgsveld[4,9] & Fabricio Loayza-Puch [1] ✉

Cellular senescence is characterized by a permanent growth arrest and is associated with tissue aging and cancer. Senescent cells secrete a number of different cytokines referred to as the senescence-associated secretory phenotype (SASP), which impacts the surrounding tissue and immune response. Here, we find that senescent cells exhibit higher rates of protein synthesis compared to proliferating cells and identify eIF5A as a crucial regulator of this process. Polyamine metabolism and hypusination of eIF5A play a pivotal role in sustaining elevated levels of protein synthesis in senescent cells. Mechanistically, we identify a p53-dependent program in senescent cells that maintains hypusination levels of eIF5A. Finally, we demonstrate that functional eIF5A is required for synthesizing mitochondrial ribosomal proteins and monitoring the immune clearance of premalignant senescent cells in vivo. Our findings establish an important role of protein synthesis during cellular senescence and suggest a link between eIF5A, polyamine metabolism, and senescence immune surveillance.

Senescence is a cellular stress program that inhibits the replication of aging and damaged cells. It can be triggered by various factors, such as telomere attrition, DNA damage, and oncogene activation[1]. One of the prominent characteristics of cellular senescence is the acquisition of a flat and enlarged morphology, along with increased expression of senescence-associated β-galactosidase (SA-β-gal). Senescent cells exhibit a loss of replicative capacity due to the activation of pathways involving p53

and p16[INK4A]-RB. Inactivation of p53 is sufficient to bypass Ras[V12]-induced senescence[2,3].

Senescent cells release a variety of cytokines collectively known as the senescence-associated secretory phenotype (SASP), which impact the surrounding tissue and immune response[1,4]. During the early stages of tumorigenesis, the SASP plays primarily a protective role by stimulating the immune system to eliminate pre-malignant cells[5]. However, prolonged exposure to the SASP is associated with chronic

[1]Translational Control and Metabolism, German Cancer Research Center (DKFZ), Heidelberg, Germany, Heidelberg, Germany. [2]Department of Thoracic Oncology, Tianjin Medical University Cancer Institute & Hospital, National Clinical Research Center for Cancer, Key Laboratory of Cancer Prevention and Therapy, Tianjin, Tianjin's Clinical Research Center for Cancer, Tianjin, China. [3]Faculty of Biosciences, University of Heidelberg, Heidelberg, Germany. [4]Proteomics of Stem Cells and Cancer, German Cancer Research Center (DKFZ), Heidelberg, Germany. [5]Division of Cell Signaling and Metabolism, German Cancer Research Center (DKFZ), Heidelberg, Germany. [6]Division of Cell Biology, Metabolism and Cancer, Department Biomolecular Health Sciences, Faculty of Veterinary Medicine, Utrecht University, CL Utrecht, The Netherlands. [7]Cell Plasticity and Epigenetic Remodeling, German Cancer Research Center (DKFZ), Heidelberg, Germany. [8]Institute of Pathology, University Hospital Heidelberg, Heidelberg, Germany. [9]Medical Faculty, University of Heidelberg, Heidelberg, Germany. [10]These authors contributed equally: Xiangli Jiang, Ali Hyder Baig. ✉e-mail: f.loayza-puch@dkfz-heidelberg.de

inflammation, malignant transformation of neighboring cells, and re-entry of cancer senescent cells into the cell cycle[6].

The eukaryotic translation initiation factor 5 A (eIF5A) is a highly conserved protein that promotes the translation of specific mRNAs encoding tri-peptide motifs rich in proline, glycine, and charged amino acids[7–10]. eIF5A has been implicated in cancer progression, chemo-resistance, and metastasis, underscoring its significance in tumor biology. Notably, eIF5A undergoes a unique post-translational modification known as hypusination, which is crucial for its activity[11]. This modification is essential for eIF5A's function and strictly relies on the naturally occurring polyamine spermidine as a substrate. Spermidine has been shown to extend the lifespan of multicellular organisms. In mice, it has also been demonstrated to delay the onset of age-related disorders[12]. Consistently, eIF5A hypusination decreases with aging in vivo and can be restored by supplementation of spermidine[13,14]. Recent evidence has shown a role for eIF5A in regulating mitochondrial homeostasis[15,16]; however, our understanding of how the hypusination of eIF5A acts during cellular senescence remains incomplete.

Protein synthesis regulation is critical for the senescent phenotype, with the mTOR pathway being a central activator in this process. It has been reported to increase the translation of specific mRNA subsets, supporting the production of high levels of SASP[17,18]. Increased protein synthesis during senescence has also been observed in other models, including therapy-induced senescence[19–21]. Recent evidence highlights eIF5A's role in ribosome-associated quality control (RQC), a process closely linked to proteotoxic stress in aging[22,23]. However, the regulation of protein synthesis rates in senescent cells

remains poorly understood. In this study, we identify eIF5A as critically required to sustain elevated rates of protein synthesis during senescence. We discover a p53-dependent program in senescent cells that maintains spermidine levels and hypusination of eIF5A. Finally, we demonstrate that eIF5A activity is necessary for the translation of mitochondrial ribosomal proteins and immune surveillance of senescent cells in vivo.

## Results

### Protein synthesis is elevated in senescent cells

To study the rate of protein synthesis in senescent cells, we used a fluorogenic assay based on O-propargyl-puromycin (OP-Puro), an alkyne analog of puromycin. Like puromycin, OP-Puro forms covalent bonds with nascent polypeptide chains within cells, and can be fluorescently labeled via click chemistry, allowing for quantification of protein synthesis by flow cytometry at the individual cell level[24] (Fig. 1a). Fibroblasts treated with OP-Puro showed a noticeable increase in fluorescence compared to cells treated with PBS. Importantly, treatment with the protein synthesis inhibitor cycloheximide (CHX) inhibited OP-Puro incorporation (Supplementary Fig. 1a).

We adapted this approach to quantify protein synthesis in primary human BJ fibroblasts that expressed hTERT and 4-OH-tamoxifen (4-OHT)-inducible oncogenic H-Ras$^{V12}$ (BJ-Ras-ER cells)[25]. These cells undergo irreversible senescence after twelve days of H-Ras$^{V12}$ activation (Supplementary Fig. 1b–d). Senescent BJ-Ras-ER cells exhibited significantly higher OP-Puro incorporation than proliferating cells (Fig. 1b), suggesting that senescent cells synthesize more protein per

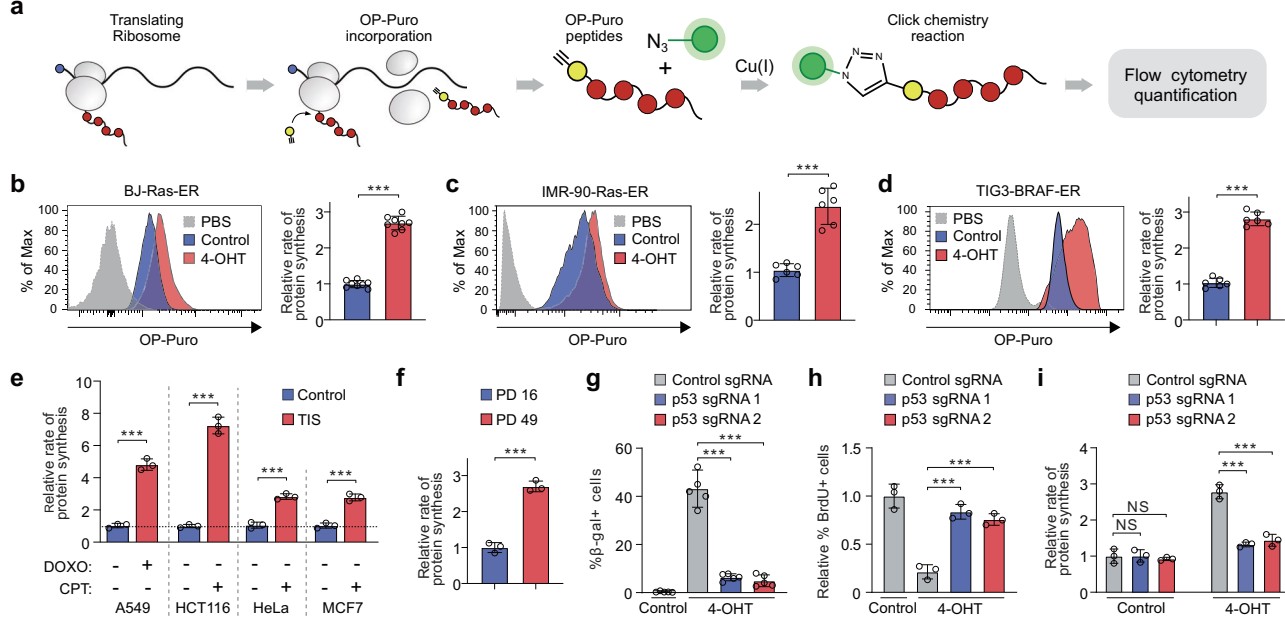

**Fig. 1 | Global translation rates are elevated in senescent cells. a** Experimental approach to measuring global translation rates in senescent cells. **b–d** O-propargyl-puromycin (OP-Puro) incorporation and protein synthesis rates in proliferating and senescent BJ-Ras-ER cells (**b**), IMR-90-Ras-ER cells (**c**), and TIG3-BRAF-ER cells (**d**). Data represent mean ± SD from biologically independent experiments (**b**, $n = 8$; **c**, **d**, $n = 6$). $p$-values were calculated using a two-tailed unpaired $t$ test. ***$P < 0.001$. **e** Protein synthesis rates in proliferating and senescent cells treated with doxorubicin (DOXO) or camptothecin (CPT) for 7 days. A549, HCT116, HeLa, and MCF7 cells were used. Data represent mean ± SD from biologically independent experiments ($n = 3$). $p$-values were calculated using a two-tailed unpaired $t$ test. ***$P < 0.001$. **f** OP-Puro incorporation assay in IMR-90 cells undergoing replicative senescence. PD, population doubling. Data represent mean ± SD from biologically independent experiments ($n = 3$). $p$-values were calculated using a two-tailed unpaired $t$ test. ***$P < 0.001$. **g** Quantification of senescence-associated beta-

galactosidase (SA-β-gal) expression in BJ-Ras-ER cells transduced with sgRNAs targeting the p53 gene or a control sequence. Data represent mean ± SD from biologically independent experiments ($n = 5$). $p$-values were calculated using a two-tailed unpaired $t$ test. ***$P < 0.001$. **h** Quantification of bromodeoxyuridine (BrdU) incorporation in various CRISPR-Cas9-transduced BJ-Ras-ER cells. Data represent mean ± SD from biologically independent experiments ($n = 3$). $p$-values were calculated using a two-tailed unpaired $t$ test. ***$P < 0.001$. **i** Quantification of protein synthesis rates based on OP-Puro incorporation in proliferating and senescent BJ-Ras-ER cells transduced with either a control sgRNA or sgRNAs targeting the p53 gene. Data represent mean ± SD from biologically independent experiments ($n = 3$). $p$-values were calculated using a two-tailed unpaired $t$ test. NS, not significant; ***$P < 0.001$. Source data, including exact $p$-values, are provided as Source Data files.

hour than their dividing counterparts. We observed a similar increase in protein synthesis in other models of oncogene-induced senescence: IMR-90 human primary fibroblasts with a 4-OHT-inducible ER-Ras oncogene (Fig. 1c) and TIG3 human fibroblasts expressing a constitutively activated form of the BRAF (V600E) fused to the estrogen receptor (TIG3-BRAF-ER) (Fig. 1d). This effect was not unique to oncogene-induced senescence. Some cancer cells undergo senescence upon treatment with drugs that induce DNA double-strand breaks, a process known as therapy-induced senescence (TIS). A549, HCT116, HeLa, and MCF7 cells treated with doxorubicin or camptothecin undergo TIS after 7 days of treatment[26] (Supplementary Fig. 1e, f). TIS cells exhibited several-fold higher OP-Puro incorporation rates compared to proliferating controls (Fig. 1e). Furthermore, replicative senescence in wild-type IMR-90 cells (population doubling (PD) 49) also led to increased rates of OP-Puro incorporation (Fig. 1f), suggesting that elevated protein synthesis is a general feature of senescent cells.

Oncogene activation promotes nearly all stages of mRNA translation and may affect protein synthesis rates[27]. To determine whether elevated OP-Puro incorporation in OIS cells is independent of oncogenic H-Ras[V12] expression, we transduced BJ-Ras-ER cells with sgRNAs against p53. After oncogenic H-Ras[V12] expression, p53 knockout cells overcame senescence and became transformed (Fig. 1g, h)[25]. OP-Puro assay showed that transformed cells had reduced protein synthesis rates compared to senescent cells (Fig. 1i). Altogether, our data indicate that senescent cells significantly increase protein synthesis rates independently of oncogene activation.

## eIF5A is required to sustain elevated rates of protein synthesis during cellular senescence

To identify genes that sustain elevated rates of protein synthesis during oncogene-induced senescence (OIS), we designed a fluorescence-activated cell sorting (FACS)-based CRISPR-Cas9 screening strategy that uses OP-Puro incorporation as a readout (Fig. 2a). To this end, we pooled 308 single guide RNAs (sgRNAs)-containing lentiviruses targeting all known genes encoding translation factors in humans (70 genes, 4 sgRNAs per gene and 28 negative controls; Supplementary Fig. 2a) and then transduced this library in BJ-Ras-ER cells. After selection with puromycin, we treated the cells with 4-OHT (senescence induction) or vehicle and subjected them to OP-Puro incorporation, Click chemistry, and FACS sorting (Fig. 2a). We reasoned that sgRNAs negatively affecting protein synthesis would be enriched in the cell population with low OP-Puro incorporation. Therefore, we harvested the cells in this fraction, isolated genomic DNA, amplified integrated vectors by polymerase chain reaction (PCR), and used next-generation sequencing (NGS) to quantify the abundance of integrated sgRNAs present in the proliferating and senescent (4-OHT) populations (Fig. 2a).

Among the sgRNAs that were selectively enriched in the low-OP-Puro population (Fig. 2b), *eIF5A* sgRNAs were the most significantly enriched in senescent but not in proliferating cells (Fig. 2b and Supplementary Data 1). eIF5A is involved in protein synthesis, cancer progression, and chemo-resistance[28]. Activated eIF5A binds to the E-site of stalled ribosomes, where it interacts with the peptidyl-tRNA situated at the P-site and promotes elongation of selected tri-peptide sequences[7,8,29]. To validate the results of our screen, we selected two independent sgRNAs against *eIF5A* (Fig. 2c) and tested their effect on OP-Puro incorporation in senescent and proliferating cells. sgRNAs targeting *eIF5A* were able to significantly decrease the rate of protein synthesis in OIS cells while proliferating control cells did not exhibit changes (Fig. 2d). *eIF5A* knockout did not affect cell growth arrest in senescent cells (Fig. 2e). Senescence-associated (SA)-β-galactosidase activity and upregulation of p16[INK4A] (encoded by *CDKN2A*) and p21[CIP1] (encoded by *CDKN1A*) mRNAs were not affected upon eIF5A depletion during OIS (Fig. 2f, g). Importantly, we observed the same effect in

therapy-induced senescent A549 cells expressing *eIF5A* sgRNAs (Supplementary Fig. 2b–d).

Next, to determine whether eIF5A depletion affects protein synthesis and growth arrest in already senescent cells, we transfected siRNAs against *eIF5A* at day 8 after 4-OHT treatment (Fig. 2h, i). Knocking down eIF5A in already senescent cells also reduced OP-Puro incorporation in senescent cells without reverting the growth arrest (Fig. 2j, k), indicating that eIF5A depletion prevents elevated rates of protein synthesis once senescence has been established. Taken together, our data reveal that eIF5A sustains high rates of protein synthesis in senescent cells without affecting growth arrest or transcription of key components of the senescence program.

## p53 modulates the expression of components of the polyamine synthesis pathway during cellular senescence

eIF5A is the only known protein to carry the post-translational modification hypusination[30]. This modification is essential for the activity of eIF5A, highly conserved among archaea and eukaryotes, and closely linked to arginine and polyamine metabolism (Fig. 3a). In human cells, the polyamine spermidine is the substrate for the hypusination of a conserved lysine residue in eIF5A, which involves two sequential steps catalyzed by deoxyhypusine synthase (DHPS) and deoxyhypusine hydroxylase (DOHH)[28] (Fig. 3a). OIS BJ-Ras-ER cells treated with N1-guanyl-1,7-diaminoheptane (GC7), a DHPS inhibitor[31], were more sensitive than proliferating cells in sustaining hypusination levels over time (Fig. 3b, c). Similarly, DHPS- or DOHH-deficient OIS cells showed a reduction in eIF5A hypusination levels (Supplementary Fig. 3a). Consistent with these observations, GC7 treatment or DHPS/DOHH knockout reduced protein synthesis rates exclusively in senescent cells (Fig. 3d and Supplementary Fig. 3b), suggesting a role for polyamine metabolism in the regulation of elevated rates of protein synthesis in senescent cells.

The first step in polyamine synthesis is the transformation of ornithine into putrescine, which is catalyzed by Ornithine Decarboxylase 1 (ODC1). Subsequently, putrescine undergoes a sequential conversion process orchestrated by Spermidine Synthase (SRM) and Spermine Synthase (SMS), resulting in the formation of spermidine and spermine, respectively (Fig. 3a). To investigate the role of polyamines in the hypusination of eIF5A during OIS, we conducted a metabolomics analysis using LC/MS to measure the levels of polyamine metabolites in proliferating and senescent BJ-Ras-ER cells. While the intracellular levels of ornithine remained unchanged between proliferating and senescent cells, putrescine and N-acetyl-putrescine levels were significantly reduced in OIS cells (Fig. 3e). However, intracellular levels of spermidine remained similar between the two conditions, while N-acetyl-spermidine levels were increased (Fig. 3e). Consistent with the comparable levels of spermidine, hypusination of eIF5A was not affected in OIS or TIS (Fig. 3f and Supplementary Fig. 3c). These findings suggest that a compensatory mechanism to maintain spermidine and eIF5A hypusination levels might exist in senescent cells despite low levels of putrescine.

To explore this possibility, we evaluated the expression of key enzymes involved in polyamine biosynthesis or catabolism. qRT-PCR analysis revealed that senescent BJ-Ras-ER and IMR-90-Ras-ER cells exhibited increased expression of Spermidine/Spermine N1-Acetyltransferase 1 (*SAT1*) and Spermine Oxidase (*SMOX*) mRNAs, while the expression of other enzymes in the polyamine pathway remained unchanged (Fig. 3g and Supplementary Fig. 3d). Similarly, therapy-induced senescence in A549 and HCT116 cells and replicative senescence in IMR-90 cells led to increased expression of *SAT1* and *SMOX* mRNAs (Supplementary Fig. 3e, f).

Regulating polyamine levels involves the reverse conversion of spermine to spermidine and putrescine. This process can be accomplished through two metabolic pathways. First, SAT1 facilitates the acetylation of spermine, leading to the production of N1-acetyl-

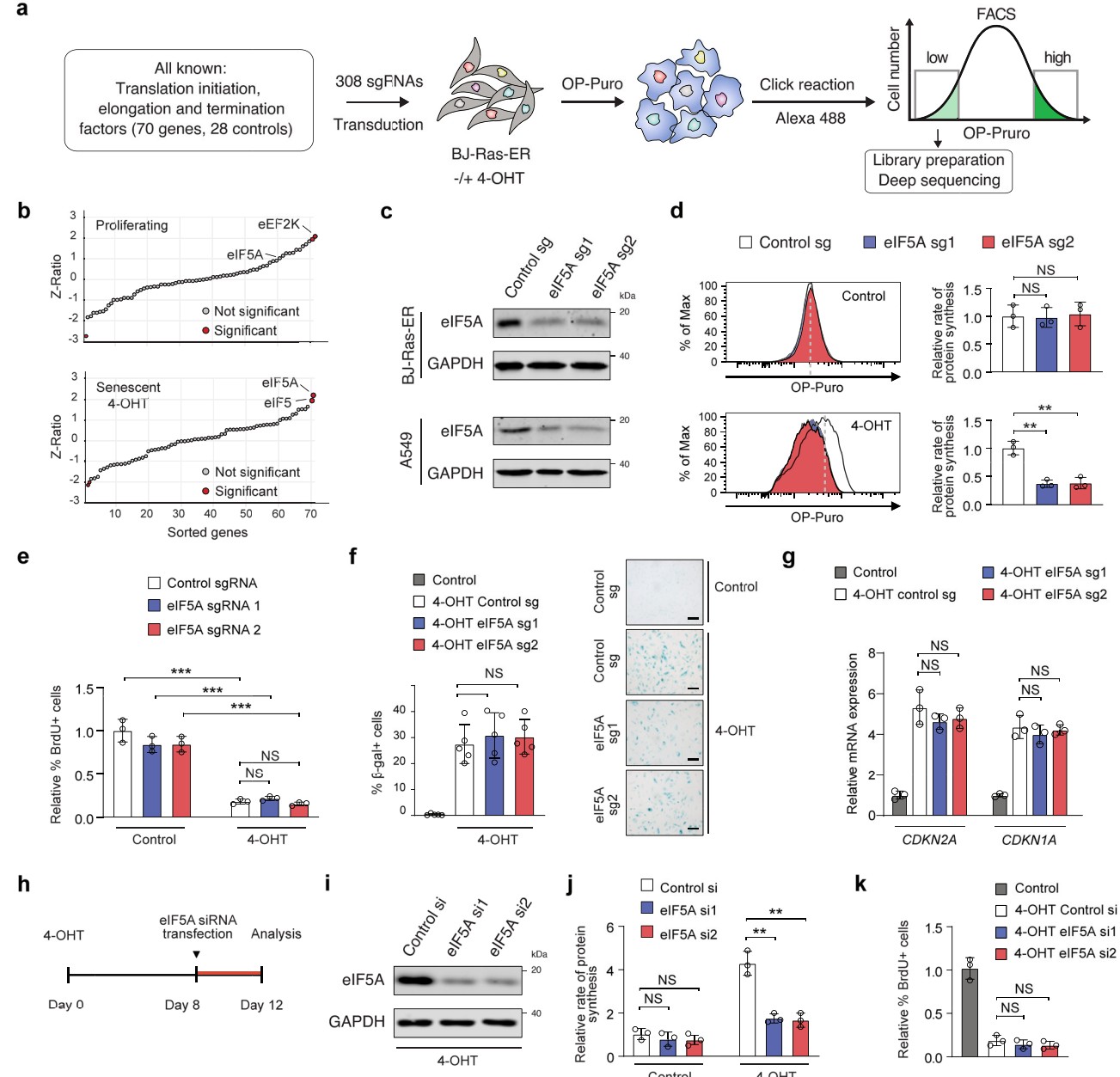

**Fig. 2 | eIF5A is required to sustain elevated protein synthesis rates during cellular senescence. a** Experimental setup for the CRISPR-Cas9 screen on protein synthesis rates. **b** CRISPR screen results, with sgRNAs sorted by enrichment in proliferating (top) and senescent cells (bottom). Z-score and Z-ratio calculations are detailed in the methods. **c** Western blots from BJ-Ras-ER (top) and A549 (bottom) cells with *eIF5A*-targeting sgRNAs, representative of 3 independent experiments. **d** OP-Puro incorporation in proliferating (top) and senescent BJ-Ras-ER cells (bottom) with control or *eIF5A* sgRNAs. Data show mean ± SD from n = 3 biologically independent experiments. *p*-values via two-tailed unpaired *t* test. NS, not significant; **P < 0.01. **e** BrdU incorporation in BJ-Ras-ER cells post 12 days of 4OHT treatment, transduced with *eIF5A*-targeting sgRNAs. Data are mean ± SD from n = 3 biologically independent experiments. *p*-values via two-tailed unpaired *t* test. NS, not significant; ***P < 0.001. **f** β-galactosidase-positive cell quantification after 12 days of 4OHT treatment, transduced with *eIF5A*-targeting sgRNAs. Mean ± SD from n = 6 biologically independent experiments. *p*-values via two-tailed unpaired *t* test. NS is not significant. Scale bars, 50 μm. **g** qRT-PCR mRNA quantification after 12 days of 4OHT treatment in BJ-Ras-ER cells with *eIF5A* sgRNAs. Data are mean ± SD from n = 3 biologically independent experiments. *p*-values via two-tailed unpaired *t* test. NS is not significant. **h** Experimental design for transfecting two siRNAs targeting *eIF5A* in BJ-Ras-ER cells. siRNAs were transfected on day 8 post 4-OHT treatment and analyzed on day 12. **i** Western blots of BJ-Ras-ER cells transfected with siRNAs against *eIF5A*, collected on day 12 post 4-OHT treatment, representative of 3 independent experiments. **j** OP-Puro incorporation quantification in proliferating and senescent BJ-Ras-ER cells transfected with siRNA control or *eIF5A* siRNAs. Data are mean ± SD from n = 3 biologically independent experiments. *p*-values via two-tailed unpaired *t* test. NS, not significant; **P < 0.01. **k** BrdU incorporation in BJ-Ras-ER cells with *eIF5A*-targeting siRNAs. Data are mean ± SD from n = 3 biologically independent experiments. *p*-values via two-tailed unpaired *t* test. NS is not significant. Source data and exact *p*-values are provided in the Source Data file.

spermine, while spermidine is transformed into N1-acetyl-spermidine. Subsequently, the enzyme polyamine oxidase (PAOX) enables the oxidation of these two products, resulting in the formation of spermidine and putrescine, respectively. Second, SMOX can restore spermidine pools by directly catalyzing the oxidation of spermine to spermidine (Fig. 3a). SAT1 has been shown to be a direct target of p53[32], and the transcriptional regulation of SMOX has been characterized only under hypoxic conditions[33]. We hypothesized that SAT1 and

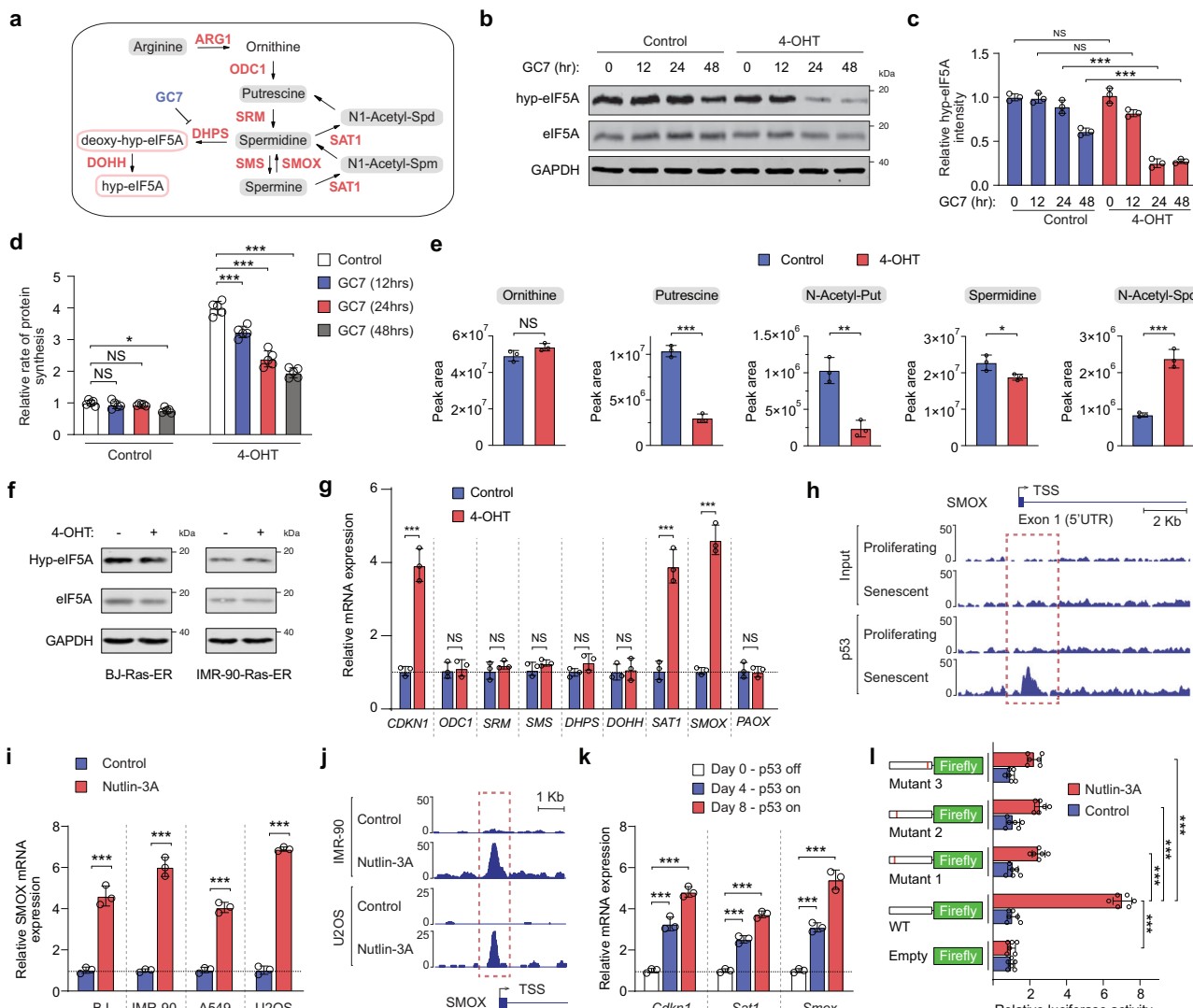

**Fig. 3 | p53 regulates the expression of polyamine pathway components during cellular senescence. a** Schematic of the polyamine pathway, showing synthesis of putrescine, spermidine, and spermine from ornithine. Spermidine is a substrate for eIF5A hypusination. **b** Immunoblotting of protein extracts from proliferating and OIS BJ-Ras-ER cells treated with GC7 (10 μM) for indicated times. **c** Hypusinated eIF5A (hyp-eIF5A) band intensity quantified relative to eIF5A. Mean ± SD from $n = 3$ biologically independent experiments. $p$-values via two-tailed unpaired $t$ test. NS, not significant; ***$P < 0.001$. **d** OP-Puro incorporation assay in proliferating and OIS BJ-Ras-ER cells treated with GC7 (10 μM). Mean ± SD from $n = 5$ biologically independent experiments. Two-tailed unpaired $t$ test $p$-values: NS, not significant; *$P < 0.05$; ***$P < 0.001$. **e** Intracellular levels of ornithine, putrescine, N-acetyl-putrescine, spermidine, spermine, and N-acetyl-spermidine measured by LC-MS in proliferating and OIS BJ-Ras-ER cells (4-OHT). Mean ± SD from $n = 3$ biologically independent experiments. Two-tailed unpaired $t$ test $p$-values: NS, not significant; *$P < 0.05$; ***$P < 0.001$. **f** Immunoblot of hypusinated eIF5A (Hyp-eIF5A) and total eIF5A in proliferating and senescent BJ-Ras-ER and IMR90-Ras-ER cells. One representative of 3 independent experiments. **g** qRT-PCR analysis of polyamine pathway gene expression in proliferating and OIS cells (4-OHT). Mean ± SD from $n = 3$ biologically independent experiments. Two-tailed unpaired $t$ test $p$-values: NS, not significant; ***$P < 0.001$. **h** *SMOX* promoter locus overview with p53 ChIP-seq data in proliferating and OIS IMR-90 cells (~10 kb). Data from Kirschner et al. **i** qRT-PCR of *SMOX* expression after 24 hrs of Nutlin3A treatment. Mean ± SD from $n = 3$ biologically independent experiments. Two-tailed unpaired $t$ test $p$-values: ***$P < 0.001$. **j** *SMOX* promoter locus schematic showing p53 binding by ChIP-seq in IMR-90 and U2OS cells after 24 hrs Nutlin-3A treatment (~5 kb). **k** Mouse liver progenitor cells with H-RasG12D, tTA, and tetracycline-responsive p53 shRNA, cultured without doxycycline to restore p53 expression. *Sat1* and *Smox* expression were assessed by qRT-PCR on days 0, 4, and 8. Mean ± SD from $n = 3$ biologically independent experiments. Two-tailed unpaired $t$ test $p$-values: ***$P < 0.001$. **l** Luciferase activity measured in A549 after 24 hrs treatment with vehicle or Nutlin3A. Mean ± SD from $n = 6$ biologically independent experiments. Two-tailed unpaired $t$ test $p$-values: ***$P < 0.001$. Source data, including exact $p$-values, are in the Source Data file.

SMOX might be directly regulated by p53 to recycle spermine back to spermidine and sustain eIF5A hypusination levels in senescent cells. To test this, we analyzed ChIP-seq data and found that p53 binds directly to the *SMOX* promoter in IMR-90 senescent cells (Fig. 3h). Further, to test whether p53 activation directly regulates *SMOX*, we stabilized the expression of the endogenous p53 gene in several cell lines using Nutlin-3A, a specific p53 activator. Treatment with Nutlin-3A resulted in increased expression of *SMOX* mRNA in BJ, IMR-90, A549, and U2OS cells (Fig. 3i) and increased p53 binding to the *SMOX* promoter in IMR-90 and U2OS cells (Fig. 3j). In addition, p53 restoration induced *Smox* and *Sat1* mRNA expression in murine liver progenitor cells expressing oncogenic Ras (H-Ras^G12D) (Fig. 3k). We also observed that depletion of p53 in BJ cells or in the isogenic pair of HCT116 WT (p53 + /+) and HCT116 p53 − /− cells reduced *SMOX* mRNA expression (Supplementary Fig. 3g, h). Finally, we cloned the *SMOX* promoter, which spanned putative p53 response elements (TSS + 785, TSS − 588, and Supplementary Fig. 3i), into a luciferase reporter gene and observed robust activation upon p53 stabilization (Fig. 3l). Mutation of the p53

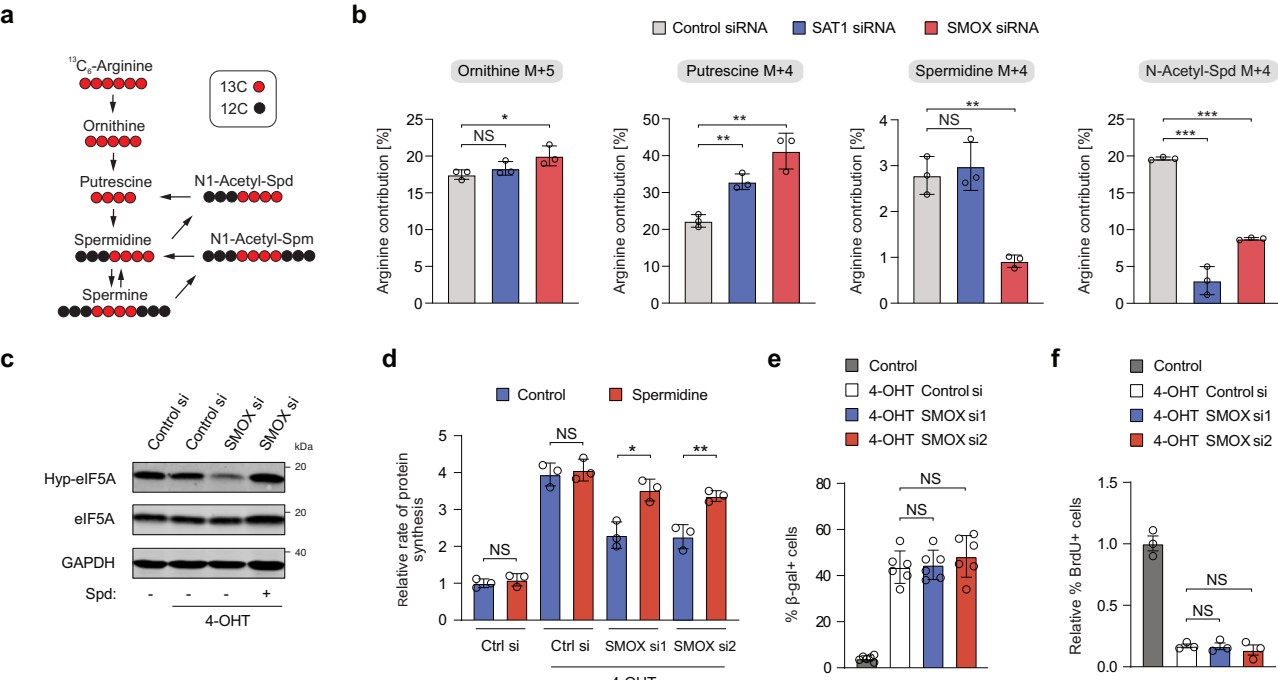

**Fig. 4 | SMOX plays a crucial role in maintaining eIF5A hypusination and protein synthesis in OIS. a** Schematic for $^{13}C_6$-Arginine tracing into the polyamine synthesis pathway. Black circles represent $^{12}C$ atoms, while red circles represent $^{13}C$ atoms. **b** Tracing analysis from $^{13}C_6$-Arginine OIS cells transfected with siRNAs targeting *SAT1* or *SMOX*. Data represent mean ± SD from biologically independent experiments ($n = 3$). *p*-values were calculated using a two-tailed unpaired *t* test. NS, not significant; *$P < 0.05$; **$P < 0.01$; ***$P < 0.001$. **c** Immunoblot assay of hypusinated eIF5A (Hyp-eIF5A), total eIF5A, and SMOX in OIS cells transfected with siRNAs against *SMOX* and treated with 10 µM spermidine (Spd). **d** Quantification of OP-Puro incorporation in OIS BJ-Ras-ER cells transfected with either siRNA control or *SMOX* siRNAs. Cells were treated with 10 µM spermidine. Data represent mean ± SD

from biologically independent experiments ($n = 3$). *p*-values were calculated using a two-tailed unpaired *t* test. NS is not significant; *$P < 0.05$; **$P < 0.01$. **e** Senescence-associated beta-galactosidase (SA-β-gal) assay in OIS BJ-Ras-ER cells transfected with siRNAs against *SMOX*. Data represent mean ± SD from biologically independent experiments ($n = 3$). *p*-values were calculated using a two-tailed unpaired *t* test. NS is not significant. **f** Quantification of BrdU incorporation in OIS BJ-Ras-ER cells transfected with siRNAs against *SMOX*. Data represent mean ± SD from biologically independent experiments ($n = 3$). *p*-values were calculated using a two-tailed unpaired *t* test. NS is not significant. Source data, including exact *p*-values, are provided as Source Data files.

response elements resulted in decreased activation. Taken together, our results demonstrate that *SAT1* and *SMOX* are direct targets of p53 and suggest a role for both enzymes in sustaining the levels of spermidine and eIF5A activity during cellular senescence.

## SMOX sustains eIF5A hypusination and promotes protein synthesis during OIS

Arginine is a major substrate for the synthesis of the polyamine precursor ornithine (Fig. 4a).

We hypothesized that the p53 targets *SAT1* and *SMOX* may have a role in rewiring the polyamine pathway to sustain spermidine levels and promote protein synthesis during cellular senescence. To test this hypothesis, we performed stable isotope tracing with $^{13}C$ arginine in senescent BJ-Ras-ER cells transfected with siRNAs against either *SAT1* or *SMOX* (Supplementary Fig. 4a). Arginine accumulated similarly in ornithine in all conditions but showed an increase in putrescine after *SAT1* or *SMOX* knockdown (Fig. 4b). Importantly, spermidine levels derived from $^{13}C$ arginine was not affected after *SAT1* knockdown, but were significantly depleted after *SMOX* down-regulation, suggesting that *SMOX* expression is necessary to sustain spermidine levels in senescent cells (Fig. 4b). In line with this observation, the levels of N-acetyl-spermidine derived from $^{13}C$ arginine and total spermine levels were also decreased after *SMOX* downregulation (Fig. 4b and Supplementary Fig. 4c).

To assess the effect of SMOX on the hypusination of eIF5A, we either depleted it with siRNAs (Supplementary Fig. 4a, b) or inhibited it pharmacologically with MDL 72527, a widely used spermine oxidase inhibitor[34]. Reduced eIF5A hypusination was observed upon

knockdown of *SMOX* in senescent BJ-Ras-ER cells, while supplementation with spermidine rescued this effect (Fig. 4c). Accordingly, protein synthesis rates decreased after *SMOX* knock-down or pharmacological inhibition as measured by OP-Puro incorporation (Fig. 4d and Supplementary Fig. 4d, e), without affecting cell growth arrest (Fig. 4e, f and Supplementary Fig. 4f, g). These findings are in accordance with the notion that the p53-target *SMOX* is necessary to sustain hypusination levels and elevated protein synthesis rates in OIS cells.

## eIF5A modulates the translation of mitochondrial ribosomal proteins in senescent cells

To assess how eIF5A activity regulates protein synthesis in senescent cells, we performed global proteome analysis in proliferating and OIS BJ-Ras-ER cells treated with GC7. Inhibition of eIF5A hypusination did not significantly affect protein expression in proliferating cells (Fig. 5a), but downregulated numerous proteins in senescent cells (Fig. 5a and Supplementary Datas 2 and 3). More than half of the downregulated proteins (63%) were components of the large or small subunits of the mitochondrial ribosome (Fig. 5b and Supplementary Fig. 5a). We confirmed these findings by monitoring the expression of mitochondrial ribosomal proteins in proliferating and OIS-BJ cells through western blot analysis. We found that the components of the large and small subunits of the mitochondrial ribosome were specifically repressed in senescent cells upon GC7 treatment, while the expression of cytosolic ribosomal proteins did not change in any of the conditions (Fig. 5c, d and Supplementary Fig. 5b). Furthermore, we observed that total mRNA levels of the selected mitochondrial

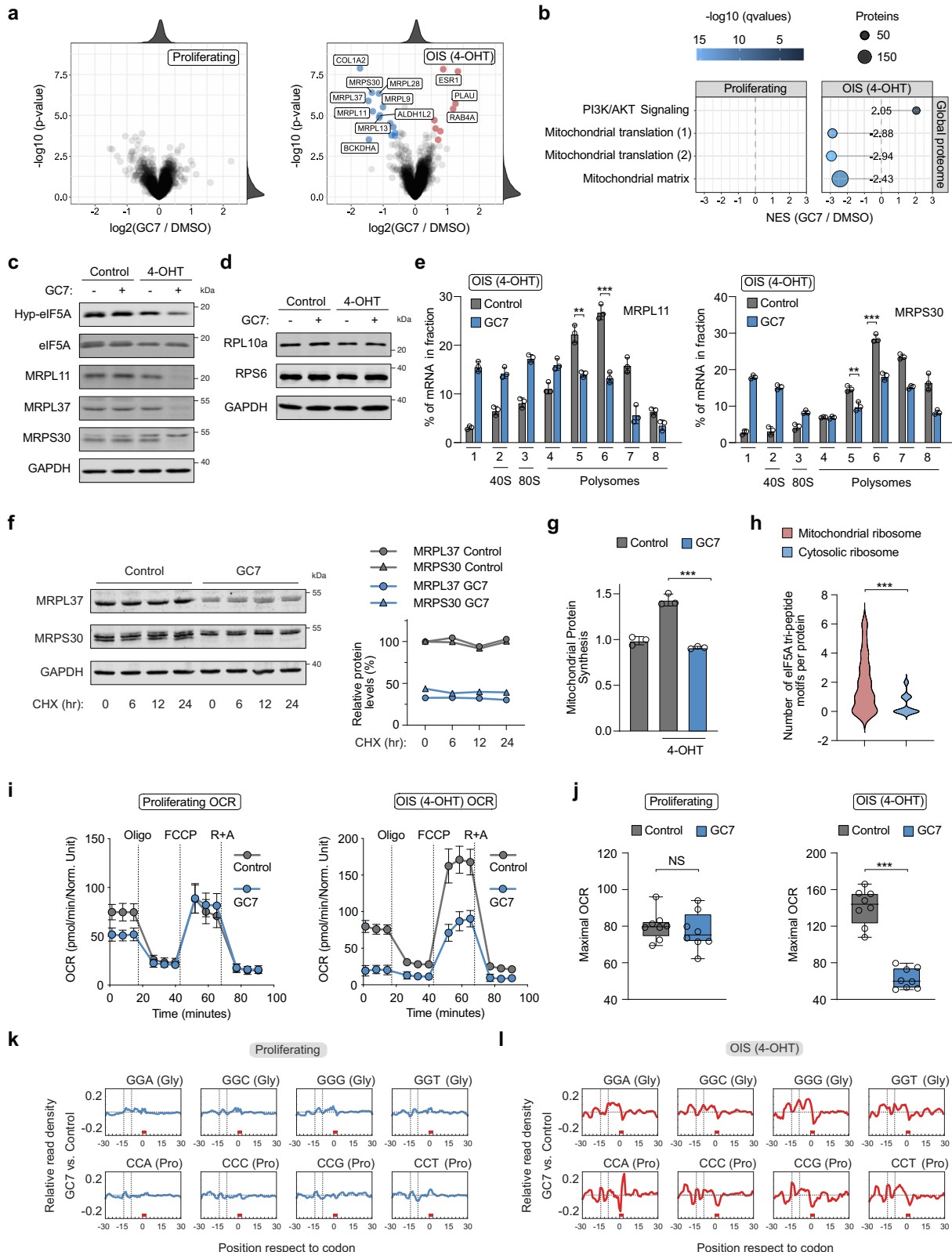

ribosomal proteins did not change upon chemical inhibition of eIF5A hypusination in proliferating and senescent cells (Supplementary Fig. 5c). However, mRNA association with polyribosomes diminished only in senescent cells exposed to GC7 (Fig. 5e and Supplementary Fig. 5d). This change could not be attributed to protein degradation (Fig. 5f). These data suggest that eIF5A plays a critical role in regulating the translation of mitochondrial ribosomal proteins in senescent cells.

In humans, the mitochondrial genome encodes 13 membrane proteins that are vital components of the electron transport chain and OXPHOS complexes, essential for cellular energy metabolism[35]. Given the dependence of mitochondrial ribosomal protein expression on eIF5A activity, it is conceivable that inhibition of eIF5A hypusination would also decrease the translation of mitochondrially encoded proteins by the mitochondrial ribosome. To test this possibility, we

**Fig. 5 | eIF5A regulates the translation of mitochondrial ribosomal proteins in OIS cells. a** Differential protein expression after GC7 addition (12 hr) in proliferating (left) and OIS-BJ cells (right). Proteins with log2 fold difference > 0.5 and adj. *p*-value < 0.001 are in red; log2 fold difference <− 0.5 and adj. *p*-value < 0.001 are in blue. Empirical Bayes moderated *t* test and two-sided *p*-values. **b** Gene set enrichment analysis from (**a**). A negative NES indicates decreased expression upon GC7 treatment. Enriched gene sets (*q*-value < 0.05) from the Molecular Signatures Database are shown, including set size and *q*-value. PI3K/AKT Signaling (R-HAS-2219528); Mitochondrial translation (1) (R-HAS-5368287); Mitochondrial translation (2) (GO:0032543); Mitochondrial matrix (GO:0005759). **c, d** Western blots on cell extracts from proliferating and OIS BJ-Ras-ER cells treated with GC7 (10 μM) for 24 h. **e** Polysome analysis in OIS BJ-Ras-ER cells treated with GC7 (10 μM) for 24 h. Mean ± SD from *n* = 3 biologically independent experiments. Two-tailed unpaired *t* test: **\*P < 0.01; \*\*\*P < 0.001. **f** Cycloheximide (CHX) chase assay in OIS BJ-Ras-ER cells treated with GC7 (10 μM) for 24 h, with quantification. Data from two independent experiments. **g** Mitochondrial OP-Puro incorporation in proliferating and

OIS BJ-Ras-ER cells treated with GC7 (10 μM) for 24 h. Mean ± SD from *n* = 3 biologically independent experiments. Two-tailed unpaired *t* test: ***P < 0.001. **h** eIF5A tri-peptide motifs per protein in human mitochondrial ribosomal proteins (*n* = 79) and all human ribosomal proteins (*n* = 76). Data represent motif distribution frequency. Chi-square test: ***P < 0.001. **i, j** Oxygen Consumption Rate (OCR) in proliferating and OIS BJ-Ras-ER cells treated with GC7 (10 μM) for 16 hours. (i) Mean ± SD from *n* = 8 biologically independent experiments. **j** Box plots (*n* = 8). The center line represents the median, upper and lower bounds represent the 75th and the 25th percentile, respectively. Whiskers represent minimum and maximum values. Oligomycin (Oligo, 0.5 μM), FCCP (1 μM), or Rotenone and Antimycin A (R + A, 0.5 μM each Two-tailed unpaired *t* test: NS is not significant; ***P < 0.001. **k, l** Cytosolic RPF density analysis in proliferating or OIS BJ-Ras-ER cells treated with GC7 (10 μM) for 18 hours. Codon regions of 61 nucleotides along the transcriptome are shown. Normalized 5′ ends of RPFs counted for each codon-region. Source data, including exact *p*-values, are provided as a Source Data file.

measured global rates of mitochondrial translation separately from cytosolic translation[36]. For this purpose, we treated cells with OP-Puro, which is incorporated into mitochondrial nascent polypeptide chains, along with MitoTracker Deep Red. We then isolated mitochondria, conjugated OP-Puro to a fluorochrome via Click chemistry, and measured both signals using flow cytometry (Supplementary Fig. 5e). We observed that the mitochondria of senescent cells had a higher rate of mitochondrial translation than those of proliferating cells; however, this effect was significantly decreased after inhibition of eIF5A hypusination (Fig. 5g). Next, we utilized a metabolic labeling approach employing pulsed stable isotope labeling (pSILAC) and L-azidohomoalanine (AHA)-based labeling of newly synthesized proteins, followed by Click-chemistry-based enrichment of labeled proteins[37]. Control and GC7-treated OIS BJ-Ras-ER cells were simultaneously labeled with AHA and either heavy or intermediate SILAC medium. Liquid chromatography followed by tandem mass spectrometry (LC-MS/MS) was used to quantify newly synthesized proteins. Our results show that the synthesis of all detected mitochondrially encoded proteins was significantly suppressed following GC7 treatment (Supplementary Fig. 5f), indicating that eIF5A activity is required for mitochondrial translation in senescent cells.

Consistent with these data, we observed decreased oxidative phosphorylation (OXPHOS) on the basis of the oxygen consumption rate (OCR) exclusively in OIS BJ-Ras-ER cells treated with GC7 (Fig. 5i, j). By contrast, aerobic glycolysis measured as extracellular acidification rate (ECAR) did not change in any of the conditions (Supplementary Fig. 5i, j). Lastly, we measured mitochondrial content through mtDNA copy number and live-cell MitoTracker staining, and observed a modest but significant increase in senescent cells, which was not altered after inhibition of eIF5A hypusination (Supplementary Fig. 5g, h).

Recent evidence showed that eIF5A acts as a global elongation factor and relieves ribosome stalling in several tripeptide motifs enriched in proline, glycine, and charged amino acids[7,8]. To study cytosolic ribosome stalling in our system, we performed ribosome profiling on proliferating and OIS BJ-Ras-ER cells treated with GC7. We performed two complementary analyses: subsequence and 5′-end density[38]. The positions 9, 12, and 15 from the 5′-end of the ribosome-protected fragments (RPFs) correspond to the E (exit), P (peptide bond), and A (tRNA recruitment and reading codon) sites of the ribosome. Both analyses revealed accumulation of RPFs at glycine and proline codons only in senescent cells treated with GC7 (Fig. 5k, l and Supplementary Fig 5k, l). Stalling at glycine codons was more prominent at site E of the ribosome, while stalling at proline codons was more pronounced at site A. Importantly, we observed accumulation of RPFs upstream of the stalling sites, indicating ribosome accumulation after inhibition of eIF5A hypusination. In line with this observation, the subset of mitochondrial ribosomal proteins contains a higher proportion of

unfavorable tripeptide motifs than the subset of cytosolic ribosomal proteins (Fig. 5h and Supplementary Data 4). Altogether, our data shows that eIF5A regulates mitochondrial function in senescent cells by promoting the translation of mitochondrial ribosomal proteins enriched in unfavorable tripeptide motifs.

## Inhibition of eIF5A hypusination affects the expression of the SASP and impairs immune surveillance of oncogenic senescent cells in vivo

Previous work suggests that the SASP is dependent on mitochondria[39–41]. Dysfunctional mitochondrial translation increases the production of reactive oxygen species (ROS)[42,43], which has been shown to negatively regulate the SASP[40,44]. Consistently, mitochondrial ROS increased only in OIS BJ-Ras-ER cells after inhibition of eIF5A hypusination (Supplementary Fig. 6a). Next, to study the effect of eIF5A hypusination on the expression of SASP proteins, we measured the expression of selected human cytokines and chemokines using commercially available membrane-based antibody arrays. Conditioned media from senescent cells showed a strong increase in the levels of CXCL1, SPP1, IL-8, CCL2, PTX3, MIF, and GDF-15, among other factors (Fig. 6a). However, conditioned media from OIS BJ-Ras-ER cells treated with GC7 showed a clear reduction in the levels of some of these cytokines, in particular CXCL1, IL-8, PTX3, and THBS1 (Fig. 6a, b). qRT-PCR analysis showed that eIF5A hypusination inhibition leads to decreased transcription of pro-inflammatory cytokines mRNAs (Fig. 6c). Consistently, the inhibition of *SMOX* by knock-down or by treatment with the small-molecule inhibitor MDL-72527 in already established senescent cells suppressed the pro-inflammatory SASP (Supplementary Fig. 6b–d). Since functional eIF5A is required to promote mitochondrial translation, we reasoned that inhibition of mitochondrial protein synthesis should also affect the pro-inflammatory SASP. Indeed, treatment with the mitochondrial translation inhibitors chloramphenicol and doxycycline suppressed the expression of key components of the inflammatory SASP (Supplementary Fig. 6e). Together, our data indicate that eIF5A-mediated mitochondrial translation promotes the expression of the pro-inflammatory SASP.

The SASP contains multiple inflammatory cytokines and chemokines that play a crucial role in the immune modulation and surveillance of pre-malignant oncogene-induced senescent cells during the initial stages of tumorigenesis[1,5]. After establishing that the p53 target SMOX modulates the activity of eIF5A during cellular senescence, we investigated whether both factors contribute to immune surveillance in a physiological context. To this end, we employed a well-established in vivo model for OIS[5,45]. In this system, senescence is induced in wild-type (WT) mouse livers by transposon-mediated transfer of oncogenic Nras (*NRas*[G12V]) by hydrodynamic tail vein injection (HTVI). Six days after injection, senescent hepatocytes develop in the liver, triggering the SASP and initiating immune-mediated surveillance. The clearance

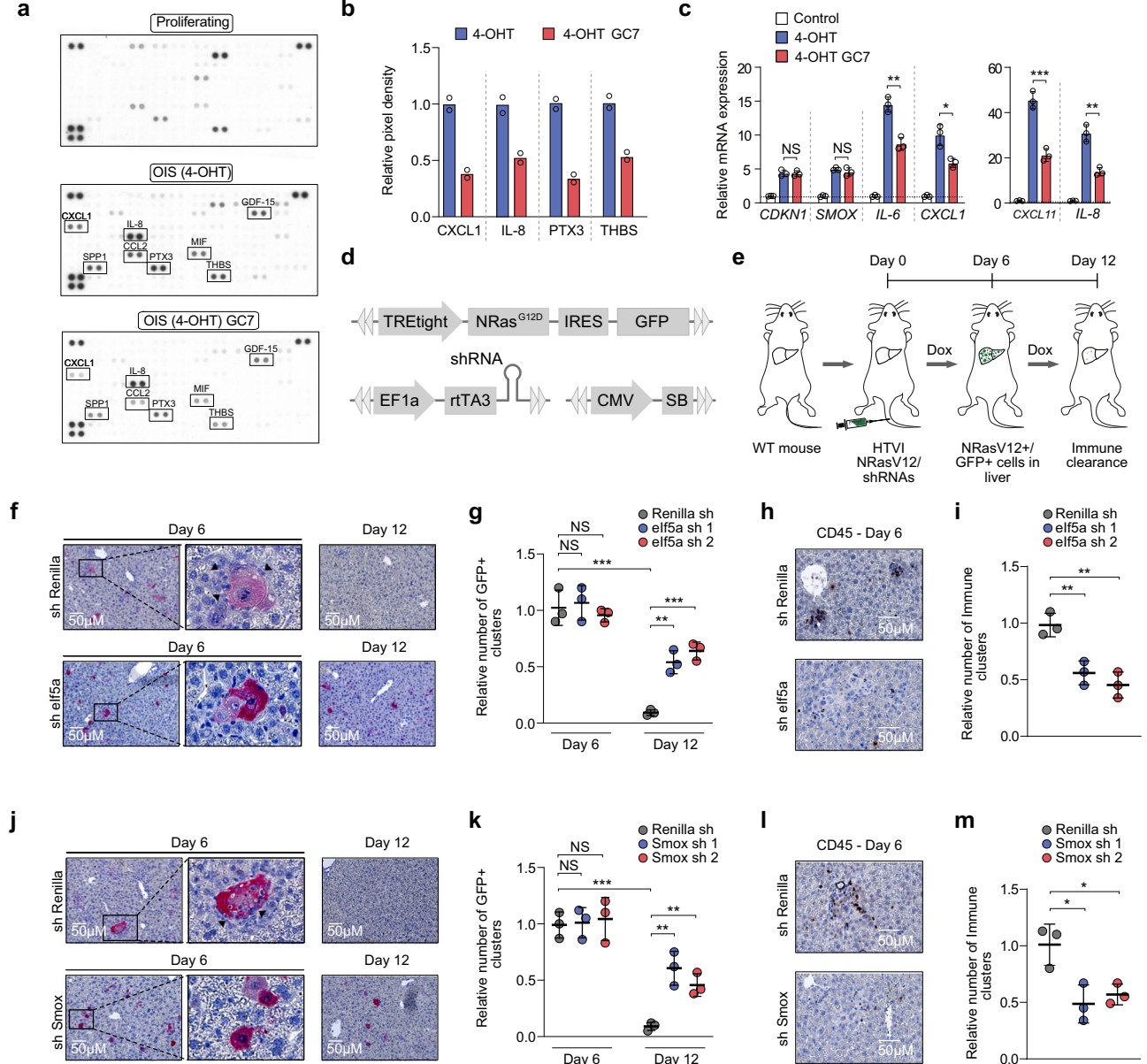

**Fig. 6 | Inhibition of eIF5A hypusination affects SASP expression, impairing immune surveillance of oncogene-induced senescent cells in vivo. a** Cytokine array of conditioned media from proliferating or OIS BJ-Ras-ER cells. OIS cells were treated with GC7 (10 µM) for 24 h. **b** Relative quantification of cytokines from (**a**). Data represent the mean of two independent experiments. **c** Expression of SASP genes in proliferating and OIS BJ-Ras-ER cells treated with GC7 (10 µM) for 48 h. Data represent mean ± SD from biologically independent experiments ($n = 3$). Two-tailed unpaired $t$ test: NS, not significant; *$P < 0.05$; **$P < 0.01$; ***$P < 0.001$. **d** Scheme of Sleeping Beauty vectors used in experiments described in (**e**). **e** Experimental design to assess effects of repressing *elf5a* and *Smox* on senescent cell clearance in the liver (HTVI, Hydrodynamic tail vein injection). **f** Representative immunohistochemistry (IHC) images of liver sections stained for GFP. Livers were collected at days 6 and 12 post-injection with vectors (**d**). Arrows indicate immune infiltrates. **g** Quantification of GFP-expressing cells in livers of animals injected with vectors. Livers were collected on days 6 and 12 post-injection. Data represent mean ± SD from biologically independent experiments ($n = 3$). Two-tailed unpaired

$t$ test: NS, not significant; **$P < 0.01$; ***$P < 0.001$. **h** Representative IHC images of liver sections stained for CD45, harvested on day 6 post-injection with vectors. **i** Quantification of infiltrated CD45-positive cells at day 6 post-injection. Data represent mean ± SD from biologically independent experiments ($n = 3$). Two-tailed unpaired $t$ test: **$P < 0.01$. **j** Representative IHC images of liver sections stained for GFP collected on days 6 and 12 post-injection with shRNAs targeting Renilla or *Smox* mRNA. Arrows indicate immune cells near senescent cell clusters. **k** Quantification of GFP-expressing cells in livers injected with Renilla or *Smox* shRNAs. Livers were collected on days 6 and 12 post-injection. Data represent mean ± SD from biologically independent experiments ($n = 3$). Two-tailed unpaired $t$ test: NS is not significant; **$P < 0.01$; ***$P < 0.001$. **l** IHC images of liver sections stained for CD45, harvested on day 6 post-injection with Renilla or *Smox* shRNAs. **m** Quantification of infiltrated CD45-positive cells at day 6 post-injection. Data represent mean ± SD from biologically independent experiments ($n = 3$). Two-tailed unpaired $t$ test: *$P < 0.05$. Source data, including exact $p$-values, are provided as a Source Data file.

process of senescent cells typically takes around 12 days. We employed this system to simultaneously express a *Nras*$^{G12D}$-IRES-GFP cassette and shRNAs targeting *elf5a* or *Smox* within the same cells. To accomplish this, we utilized two transposon vectors that were functionally linked. One vector contained a tetracycline-responsive promoter to regulate

*Nras*$^{G12D}$ expression, while the other vector constitutively expressed the reverse tet-transactivator protein (rtTA) along with a miR-30-based shRNA (Fig. 6d, e).

After six days of co-delivering both expression vectors, we observed senescent cells (GFP-positive and SA-β-gal-positive) at similar

frequencies in all conditions (Fig. 6f, g, j, k). In agreement with our in vitro findings (Fig. 2f and Fig. 4e), knockdown of *elf5a* or *Smox* did not have an effect on the number of SA-β-gal-positive cells (Supplementary Fig. 6f), indicating that senescence was induced at comparable levels in all groups. After 12 days of injection, senescent cells were cleared in the livers expressing control sh Renilla. However, knockdown of *elf5a* or *Smox* in senescent NRas^G12V-expressing hepatocytes significantly impaired immune clearance (Fig. 6f, g, j, k). This observation correlated significantly with lower numbers of infiltrated immune cells (CD45 + cells) in the sh-*elf5a*- and sh-*Smox*-expressing groups (Fig. 6h, i, l, m) compared with control groups. Taken together, these findings indicate that eIF5A and polyamine recycling in senescent cells are necessary for SASP-mediated immune clearance in vivo.

## Discussion

Senescence is a phenomenon resulting in irreversible cell growth arrest associated with aging and various diseases, including cancer. In our study, we demonstrated that OIS cells synthesize more protein than dividing cells, independent of oncogene activation. We identified the key roles of eIF5A and polyamine metabolism in sustaining elevated protein synthesis rates during oncogene-induced senescence. Notably, we discovered that SMOX, a crucial enzyme in polyamine recycling, is directly targeted by p53. Upregulation of SMOX during senescence is essential for maintaining spermidine levels, eIF5A hypusination, and enhanced protein synthesis rates. Furthermore, our research reveals that eIF5A activity plays a critical role in mitochondrial protein synthesis. Inhibition of eIF5A hypusination leads to a diminished proinflammatory senescence-associated secretory phenotype (SASP) and compromises immune surveillance in vivo.

Protein synthesis undergoes significant alterations during cellular senescence. Senescent cells undergo a state of translational reprogramming characterized by selective translation of mRNAs encoding proteins involved in senescence-associated phenotypes, such as cell cycle regulators, SASP factors, and anti-apoptotic proteins[17,18]. This reprogramming is mediated by alterations in the activity of RNA-binding proteins and microRNAs, which control mRNA translation and stability and may lead to increased protein synthesis to support the altered cellular phenotype[20,21,46]. Although further investigation into the mechanisms underlying increased protein synthesis in senescent cells is warranted to fully elucidate the role of protein synthesis dysregulation in cellular aging and age-related pathologies, our study indicates that eIF5A is required to sustain high levels of protein synthesis in senescent cells.

Polyamines play a significant role in the aging process as essential small organic molecules that affect cell growth, proliferation, and differentiation[47]. Their impact on lifespan and age-related diseases has been extensively studied, with modulation of polyamine levels influencing the aging process[12]. Spermidine, in particular, has been investigated for its anti-aging effects, extending the lifespan of multiple organisms. Spermidine enhances cellular quality control mechanisms such as autophagy, clearing toxic protein aggregates, and maintaining cellular homeostasis[48]. It also regulates translation, specifically through the modification of eIF5A, which is involved in translating specific mRNAs related to cell proliferation, autophagy, and apoptosis[11,28,49,50]. The hypusination of eIF5A, a unique post-translational modification that requires polyamines as substrates, is essential for its activity and has been implicated in aging-related processes. The polyamine-eIF5A-hypusine axis is emerging as a potential therapeutic opportunity in cancer. Recent studies show the effectiveness of DL-alpha-difluoromethylornithine (DFMO), a drug that blocks ODC and the production of polyamines, in suppressing polyamine synthesis, thereby inhibiting cell proliferation, especially in neuroblastoma and colorectal cancer[51–53].

Many components of the polyamine pathway are dynamically regulated through cell differentiation and aging[54,55]. Overexpression of

SAT1 has been suggested to facilitate polyamine recycling, while SMOX expression positively correlates with aging, presumably to sustain spermidine levels[55–57]. In our study, we observed that despite low intracellular putrescine concentrations in senescent cells, spermidine levels and eIF5A hypusination remain largely unaffected. Furthermore, we demonstrated p53's direct regulation of the polyamine pathway during cellular senescence by targeting SAT1 and SMOX. SMOX plays a crucial role in maintaining spermidine levels, eIF5A hypusination, and promoting protein synthesis in senescent cells, underscoring the significance of the polyamine pathway in the cellular senescence process (Supplementary Fig. 6g).

eIF5A participates in various aspects of mitochondrial function, including import, dynamics, membrane potential, ROS production, and potentially mitophagy, collectively contributing to the maintenance of mitochondrial integrity and cellular homeostasis[9,15]. Previous studies have shown that inhibiting eIF5A hypusination restricts mitochondrial OXPHOS in macrophages and controls the translation of specific mitochondrial proteins in a manner dependent on mitochondrial targeting signals (MTSs)[16]. In our study, we found that inhibiting eIF5A hypusination specifically downregulates several proteins, particularly the components of the large and small subunits of the mitochondrial ribosome, in senescent cells while not affecting proliferating cells. These mitochondrial ribosomal proteins are crucial for the proper functioning of the electron transport chain and oxidative phosphorylation complexes, which are essential for cellular energy metabolism. Notably, mitochondrial ribosomal proteins are enriched in tripeptide sequences that contain proline, glycine, and charged amino acids and rely on functional eIF5A for efficient translation.

The findings from our study provide evidence for the critical role of eIF5A in the regulation of mitochondrial ribosomal protein translation in senescent cells. This regulation has a significant impact on mitochondrial function and the expression of the pro-inflammatory SASP. The implications of our findings extend to the broader context of the aging process, emphasizing the importance of eIF5A and its interaction with polyamines. These insights into the molecular mechanisms underlying eIF5A's involvement in mitochondrial ribosomal protein translation shed light on potential therapeutic targets for modulating age-related diseases and enhancing immune surveillance during tumorigenesis. Further exploration of the intricate interplay between eIF5A, polyamines, and mitochondrial function may offer promising avenues for therapeutic interventions in the field of aging and cancer research.

## Methods

### Cell culture

BJ (ATCC, CRL-2522), IMR90 (ATCC, CCL-186), TIG3 (CVCL_E939), A549 (ATCC, CCL-185), HeLa (ATCC, CCL-2), MCF7 (ATCC, HTB-22), HCT116 (ATCC, CCL-247), and U2OS (ATCC, HTB-96) cells were cultured in DMEM (Gibco/Thermo Fisher Scientific, 41966) supplemented with 10% FBS and Penicillin/Streptomycin. Cells were cultivated at 37 °C with 5% CO2 in a humidified incubator. OIS cells were generated by treatment with 4-Hydroxytamoxifen (100 nM, Sigma) for 12 days. Senescence in A549 and HCT116, HeLa, or MCF7 cells was induced after a 7-day treatment with doxorubicin (100 nM) or camptothecin (20 nM), respectively. To stabilize p53, BJ-Ras-ER cells were treated with 8 μM of Nutlin-3a (Cayman Chemical) for a period of 16 hours. RNA interference experiments were performed using Dharmafect transfection reagent 1 and 25 nM of siRNA.

### OP-Puro-based quantification of protein synthesis

Cells were treated with OP-Puro (Jena Bioscience, 30 μM) for 45 min. After treatment, cells were washed with phosphate-buffered saline (PBS) and harvested by trypsinization followed by centrifugation. Cell pellets were fixed in 0.5 ml of 1% formaldehyde in PBS for 15 min on ice. After fixation, cells were washed with PBS and centrifuged. Cell pellets

were then permeabilized in 200 µl of PBS supplemented with 3% FBS and 0.5% Tween-20 (Sigma) for 5 min at room temperature. The azide-alkyne cycloaddition was carried out using the Click-iT Cell Reaction Buffer Kit (Life Technologies) and azide conjugated to Alexa Fluor 488 (Life Technologies) at a final concentration of 5 µM. Data acquisition and cell sorting were performed on FACS LSR II and FACS Aria II (BD Biosciences). Data analysis was conducted using FlowJo software.

### Senescence β-Galactosidase cell staining assay

The Senescence β-Galactosidase cell staining assay was conducted in 6-well plates using the Senescence β-Galactosidase Staining Kit (Cell Signaling Technology #9860). Initially, the growth media was aspirated, and cells were washed once with 1X PBS. Then, 1 ml of 1X Fixative Solution was added to each well and incubated for 10–15 min at room temperature. Afterward, cells were washed twice with 1X PBS, and 1 ml of the β-Galactosidase Staining Solution was added to each well. The plate was then incubated at 37 °C overnight in a dry incubator (without CO2).

### BrdU incorporation assay

Cells were treated for 1 h with 0.03 mg/mL bromodeoxyuridine (BrdU, Sigma). Following treatment, cells were washed with PBS, fixed with 70% ethanol for 5 min, and then washed two times with PBS. Subsequently, 1.5 M HCl solution was added to the cells and incubated for 30 min, followed by two additional washes with PBS. The cells were then stained with anti-BrdU (Dako) overnight at 4 °C, preceded by a 60-min blocking step with 5% Fetal Bovine Serum (FBS) in 0.3% Triton X-100 PBS. After staining, cells were washed 3 times with PBS and incubated with anti-mouse Alexa Fluor 488 secondary antibody (Dako) for 1 h at room temperature in the dark, followed by three additional washes with PBS. BrdU incorporation was assessed using immunofluorescence and High Content Analysis microscopy.

### sgRNA-mediated knockout

Four different single guide RNAs (sgRNAs) were designed against each gene. These sgRNAs were cloned into the pLentiCRISPRv2 lentiviral vector (Addgene, 52961). The knockout efficiency of the selected sgRNAs was confirmed by western blot analysis and gDNA sequencing.

### OP-Puro based CRISPR-Cas9 screen in BJ-Ras-ER cells

sgRNAs targeting all human translation factors (Supplementary Fig. 2a) were cloned into the lentiCRISPR v2-Puro vector (Addgene, 52961) at the BsmBI site. To generate viruses, HEK-293T cells were co-transfected with the following vectors: VSV-G (Addgene, 8454), pMDLg/pRRE (Addgene, 12251), pRSV-Rev (Addgene, 12253), and lentiCRISPR v2-Puro containing the sgRNAs, using JetPrime Transfection Reagent (Polyplus). The medium was replaced 4 to 6 hours after transfection. Viral particles were harvested by collecting the supernatant 24 hours after the medium change. The supernatant containing viral particles was filtered through 0.45 µm pore size filters and stored at −80 °C. For lentiviral transduction, cells were transduced with a virus-containing medium supplemented with 8 µg/mL polybrene. After 18 h, the culture medium was changed, and 24 h later, cells were selected with puromycin. Upon completion of puromycin selection, senescence was induced with 4-OHT for 12 days. Senescent and proliferating cells were treated with OP-Puro (Jena Bioscience, 30 µM) for 45 min, fixed, and stained as described earlier. Subsequently, cells were sorted, and the bottom 15% of Alexa Fluor 488-positive cells ("low OP-Puro population") were collected. The number of cells recovered was consistent with a minimum representation of 10,000 copies/sgRNA.

Enrichment scores were computed for each sgRNA in proliferating and senescent cells by contrasting its normalized frequency at day 12 (low OP-Puro population) with that at day 0 (whole population). The differences between enrichment scores (log-fold changes, LFCs) were determined for low-OP puro proliferating and senescent cells relative to proliferating cells. Subsequently, we calculated z-scores for each condition using the formula: $z = (x - \mu) / (\sigma / \sqrt{n})$, where x represents the mean residual for a gene, µ denotes the mean residual of all sgRNAs, σ signifies the standard deviation of all sgRNAs, and n stands for the number of sgRNAs for a given gene.

### Antibodies

The following antibodies and dilutions were used for Western Blot analysis: eIF5A antibody was from BD Transduction Laboratories (611977; 1:10,000), Anti-hypusine eIF5A was from EMD Millipore (ABS1064-I; 1:2,000). Anti-MRPL11 (2199; 1:500) and anti-RPS6 (2217; 1:1,000) were purchased from Cell Signaling Technologies. Anti-RPL10a was from Abcam (ab226381; 1:1000). Anti-MRPL37 (15190-1-AP; 1:500), anti MRPS30 (1844-1-AP; 1:500), anti-GAPDH (60004; 1:20,000), and anti-SMOX (15052-1-AP; 1:500) were purchased from Proteintech.

### Luciferase reporter assay

The *SMOX* promoter region was PCR amplified from the genomic DNA of BJ-Ras-ER cells. Subsequently, the PCR product was cloned into the pGL3-promoter vector (Promega). The resulting constructs were transfected into A549 cells and treated with 8 µM Nutlin-3a (Cayman Chemical). Reporter activity was assessed 36 h post-transfection using the Luciferase assay system (Promega).

### LC/MS based metabolomics and spermine quantification

BJ-Ras-ER cells were seeded in triplicate wells of 6-well plates at a density of $5 \times 10^5$ cells per well. After 24 h, the medium was replaced with DMEM containing 0.4 mM [U-$^{13}$C]-arginine (Cambridge Isotope Laboratories). Cells were harvested after 48 h. Cells were washed with ice-cold PBS and metabolites were extracted in 1 ml lysis buffer containing methanol/acetonitrile/dH2O (2:2:1). Samples were spun at 16,000 g for 15 min at 4 °C, and supernatants were collected for LC-MS analysis.

LC-MS analysis was performed on a Q-Exactive HF mass spectrometer (Thermo Scientific) coupled to a Vanquish autosampler and pump (Thermo Scientific). Polyamine metabolites were separated using an iHILIC-Fusion(P) column (2.1 × 150 mm, 5 µm, guard column 2.1 × 20 mm, 5 µm; Hilicon) using a linear gradient of methanol (A) and eluent B (75 mM (NH4)CH02, pH 4). The flow rate was set at 100 µL/min, and the gradient ran from 20% B to 80% B in 23 min, followed by a wash step and equilibration at 20% B. The MS operated in polarity-switching mode with spray voltages of 4.5 kV and − 3.5 kV. Metabolites were identified on the basis of exact mass within 5 ppm and further validated by concordance with retention times of standards. Quantification was based on peak area using TraceFinder software (Thermo Scientific), peak areas were normalized based on total signal, and isotopomer distributions were corrected for the natural abundance of $^{13}$C.

Total spermine levels were measured using the Spermine ELISA kit (FineTest).

### shRNA-mediated knockdown

Short hairpin RNAs (shRNAs) against each gene were designed using the splashRNA tool. shRNAs were cloned in the MLPE lentiviral vector. The knockdown efficiency of each shRNA was confirmed by qRT-PCR.

### Quantitative Real-time PCR analysis

Total RNA was extracted from freshly isolated and/or virally transduced cells using TRI Reagent (Zymo Research) and stored at − 80 °C until further use. cDNA synthesis was conducted using the LunaScript RT SuperMix Kit (New England Biolabs). Briefly, the reaction mixture containing 0.5–1 µg of DNase-treated RNA and 4 µl of LunaScript RT SuperMix (5X) in a total volume of 20 µl was incubated at 25 °C for

2 min. The RT reaction was then carried out at 55 °C for 10 min, followed by a termination step at 95 °C for 1 min.

Relative gene expression analysis was conducted using quantitative real-time PCR (qPCR) with the ABI QuantStudio Real-Time PCR Detection system (Applied Biosystems). qPCR was performed using the Luna Universal qPCR Master Mix (New England Biolabs) detection system. Gene expression levels were normalized to GAPDH as the housekeeping gene. Primers used are listed in Supplementary Data 5.

## Nascent- and global proteome analysis

A metabolic labeling approach, combining pulsed stable isotope-labeling (pSILAC) and L-Azidohomoalanine (AHA)-based labeling of newly-synthesized proteins, subsequent click-chemistry based enrichment of the labeled proteins, and liquid chromatography with tandem mass spectrometry (LC-MS/MS) were used for the quantitative analysis of protein synthesis. The workflow was adapted from Eichelbaum K. et al[37]. Sample preparation for global proteome analysis was carried out using SP3[58]. In brief, aggregation of proteins on the magnetic beads was induced by the addition of acetonitrile to a final concentration of 50 % (v/v) and incubating at room temperature for 18 min. The beads were subsequently washed twice with 80 % (v/v) ethanol and acetonitrile. On-bead digestion of proteins was carried out by adding 50 μL 100 mM ammonium bicarbonate (pH 8.0) and 1 μg sequencing grade modified trypsin (Promega) and incubating for 16 h at 37 °C. Following the tryptic digestion, the supernatant was removed from the beads using a magnetic rack and trifluoroacetic acid (TFA) was added to a final concentration of 1 % (v/v).

## Combined pulsed SILAC and AHA labeling

The proliferating and senescent BJ cells were treated with 10 μM GC7 or 0.1% DMSO for 5 h and 15 min and subsequently washed with warm PBS and incubated with DMEM high glucose medium deprived of Methionine, Arginine, and Lysine for 45 min. The pulsed SILAC and AHA labeling was carried out with Methionine-free DMEM high glucose medium containing heavy- ($^{13}C_6^{15}N_4$-Arg, $^{13}C_6^{15}N_2$-Lys) and intermediate ($^{13}C_6$-Arg, $D_4$-Lys) Lysine and Arginine, 10 μM GC7 or 0.1% DMSO and 100 μM AHA for 6 h. GC7-treated proliferating BJ cells were labeled using heavy, and DMSO-treated proliferating cells were labeled using an intermediate SILAC medium. GC7-treated senescent BJ cells were labeled using an intermediate, and DMSO-treated senescent cells were labeled using a heavy SILAC medium. After the total 12 h GC7 or DMSO treatment, including 6 h labeling, the cells were washed with cold PBS and the pellets were shock frozen with liquid nitrogen.

## Enrichment of newly-synthesized proteins

Cell pellets were lysed with lysis buffer containing 1% Sodium-dodecylsulfate (SDS), 300 mM HEPES (pH 8.0), and a complete EDTA-free protease inhibitor cocktail (Merck). The lysates were sonicated with a probe sonicator (Branson) at 10% power for a total duration of 1 minute. Cellular debris was removed from the lysates using centrifugation at 20,000 × *g* for 15 min and protein concentrations were determined using a BCA assay (Thermo Fischer). A total of 700 μg (350 μg per cell type) protein was used as input for the enrichment. The combined lysates were alkylated with 14.6 mM iodoacetamide (IAA) for 20 min at room temperature. Subsequently, AHA-containing proteins were coupled to 50 μL propargylamine-coupled epoxy-activated magnetic sepharose beads (Cube Biotech) via the addition of 1.15 mM $CuSO_4$, 5.77 mM Tris-hydroxypropyltriazolylmethylamine (THPTA), 11.54 mM Aminoguanidine HCl and 11.54 mM sodium ascorbate. The reaction mixture was incubated for 2 h at 40 °C. The supernatant was discarded and the beads were washed with 1.8 mL milliQ $H_2O$. Proteins bound by the beads were reduced by adding 10 mM Tris(2-carboxylethyl)phosphine (TCEP) and 40 mM 2-chloroacetamide (CAA), dissolved in 100 mM Tris-HCl buffer (pH 8.0), containing 200 mM NaCl, 0.8 mM

Ethylendiamintetraacetic acid (EDTA), 0.8% SDS and incubating at 70 °C for 20 min and subsequent incubation at 20 °C for 15 min. The beads were subsequently washed with 6 mL 1% SDS dissolved in 100 mM Tris-HCl (pH 8.0), 250 mM NaCl and 1 mM EDTA buffer, 2 mL milliQ $H_2O$, 6 mL 6 M Guanidine-HCl in 100 mM Tris-HCl (pH 8.0) and 6 mL 20% acetonitrile in milliQ $H_2O$, in 3 separate washing steps. Following the washing steps, the beads were resuspended in 200 μL 100 mM Ammonium bicarbonate buffer (pH 8.0). Proteins coupled to the beads were digested by adding 1 μg Trypsin/LysC mix (Promega) for 16 h at 37 °C. The tryptic peptides were desalted using the SP3 peptide clean-up protocol[58], dissolved in 0.1 % formic acid, and used for LC-MS/MS analysis.

## Preparation of samples for global proteome analysis

Cell lysates, which were used for the nascent proteome enrichment were also used for the preparation of global proteome analysis. 40 μg of protein, determined via BCA assay (Thermo Fisher), were used as input for the global proteome samples. The proteins were reduced by addition Tris(2-carboxylethyl)phosphine (TCEP) and 2-chloroacetaminde (CAA), to a final concentration of 10 mM and 40 mM, and incubation at 70 °C for 20 min and subsequently 15 min at 20 °C. The samples were further processed following the SP3 protocol[58]. For digestion of the proteins, the beads were resuspended in 100 mM Ammonium bicarbonate buffer (pH 8.0) and 0.8 μg Trypsin/LysC mix (Promega) was added to each sample. The tryptic peptides were desalted according to the SP3 protocol[58], dissolved in 0.1 % formic acid, and used for LC-MS/MS analysis.

## LC-MS/MS based proteomics

Quantitative measurements of tryptic peptides, of the enriched newly-synthesized proteins and global proteome, were carried out using an EASY-nLC 1200 system (Thermo Fischer Scientific) coupled to an QExactive HF mass spectrometer (Thermo Fischer Scientific). The peptides were separated in reverse-phase liquid chromatography using 0.1 % formic acid (solvent A) and 80 % acetonitrile (solvent B) as mobile phase. In order to archive separation of the peptides, they were separated on an Acclaim PepMap trap column (Thermo Fisher Scientific, C18, 20 mm × 100 μm, 5 μm C18 particles, 100 Å pore size) and a nanoEase M/Z peptide BEH C18 analytical column (Waters, 250 mm × 75 μm 1/PK, 130 Å, 1.7 μm). The samples were loaded onto the trap column with the constant flow of solvent A at a maximum pressure of 800 bar. The analytical column was equilibrated with 2 μL solvent A at a maximum pressure of 600 bar heated to 55 °C using a HotSleeve + column oven (Analytical SALES & SERVICES). The peptides were eluted with a constant flow rate of 300 nL/min. The concentration of solvent B was gradually increased during the elution of the peptides. The gradient started with 3 % solvent B for the first 4 min, increased to 8 % after 4 min, and to 10 % after 6 min. After 68 min, the percentage of solvent B was raised to 32 %, and after 86 min to 50 %. From 87 min to 94 min of the gradient, the percentage of solvent B increased to 100 %. After 95 min, the system was re-equilibrated using 3 % solvent B for 10 min. The peptides were ionized and injected, using the Nanospray flex ion source (Thermo Fischer Scientific) and a Sharp Singularity nESI emitter (ID = 20 μm, OD = 365 μm, L = 7 cm, α = 7.5°) (FOSSILIONTECH), connected to a SIMPLE LINK UNO-32 (FOSSILIONTECH). A static spray voltage of 2.5 kV was applied to the emitter and the capillary temperature of the ion transfer tube was set to 275 °C.

The QExactive HF mass spectrometer was operated in the data-dependent mode using a full scan range of 375–1500 m/z, Orbitrap resolution of 60000, automatic gain control (AGC) target of 3e6, and maximum injection time of 32 ms. Data-dependent MSMS spectra were acquired using a Top 20 scheme, using a fixed scan range from 200-2000 m/z and a fixed first mass of 110 m/z. The quadrupole isolation window was set to 2.0 m/z, and the normalized collision energy was set to 26. The Orbitrap resolution was set to 15000 with an AGC target of

1e5 and a maximum injection time of 50 ms. The data type for the MSMS spectra was set to profile mode and a charge state exclusion of 1, 5–8 & > 8 was defined. An intensity threshold of 2e4 and a minimum AGC target of 1e3 were set for the data-dependent MSMS spectra acquisition.

## Analysis of proteomic data

Raw files were processed using Maxquant version 2.0.1 and 2.0.3 and the Andromeda search engine[59]. A human proteome fasta file, retrieved from the SwissProt database (version from February 2021 with 20934 entries) was used for the analysis of the samples. The enzymatic digestion was set to Trypsin/P and a maximum of 2 missed cleavages per peptide were allowed. For the analysis of the nascent proteome, raw files of both the nascent proteome and global proteome samples were processed together, using Maxquant version 2.0.3. The multiplicity was set to 3, comprising a light channel, an intermediate channel with Arg6 and Lys4, and a heavy channel with Arg10 and Lys8. Cysteine carbamidomethylation was set as a fixed modification, whereas Methionine oxidation, N-terminal acetylation, Lysine acetylation, and deamidation of Asparagine and Glutamine were set as variable peptide modifications. The Re-quantify, match between runs, and dependent peptide search options were enabled with default parameters. Unique and razor peptides were used for quantification, and normalized SILAC ratios and iBAQ values were calculated. The minimum ratio count was set to 0 to not exclude identifications in single SILAC channels. The PSM and protein FDR threshold was set to 1%. SILAC ratios of the nascent proteome samples were median normalized to the mean SILAC ratios of the top 1000 (iBAQ) protein groups in the global proteome samples.

For the analysis of the global proteome, the raw files of the global proteome samples were carried out separately using Maxquant version 2.0.1. Cysteine carbamidomethylation was set as fixed modification, whereas Methionine oxidation, N-terminal acetylation, Lys4, Lys8, Arg6, Arg10, and substitution of Methionine to AHA were set as variable peptide modifications. Match between runs, calculation of iBAQ values, and label-free quantification were enabled with default settings.

The proteinGroups.txt output table was processed in the R software environment (version 4.0.3) using custom scripts. Protein groups with a minimum of 2 SILAC ratios or LFQ intensity values in 3 replicates were used for downstream analysis. Differential abundance testing with the proteomic data was carried out using the Limma[60] and DEqMS[61] R/Bioconductor packages. The data was fitted onto a linear model, and an empirical Bayes-moderated $t$ test was performed. The number of SILAC ratios for the nascent proteome samples and the number of identified peptides for the global proteome samples were included as a factor for the variance estimation in DEqMS. $P$-values were adjusted using the Benjamini-Hochberg approach. Geneset enrichment analysis (GSEA) of the proteomic data was carried out using the clusterProfiler R/Bioconductor package. Gene lists of the Molecular Signatures Database were retrieved and analyzed using the msigdbr package of the CRAN software repository. Gene sets of the Hallmark (H), curated gene set (C2), and Ontology gene set (C5) subcategories were included in the analysis. All quantified proteins were ordered according to the log2 fold change values and used as input for the GSEA, $p$-values were adjusted using the Benjamini-Hochberg approach.

## Oxygen consumption and extracellular rate assay

Oxygen consumption and extracellular acidification rates were measured on a Seahorse XFe96 analyzer (Agilent). Briefly, BJ-Ras-ER cells were seeded at $1 \times 10^5$ (proliferating) or $8 \times 10^4$ (senescent) cells per well in an XF96 cell culture microplate. After attachment, cells were treated with GC7 (10 μM) for 16 h. The next day, cells were equilibrated in XF assay media supplemented with 17.5 mM glucose and 2 mM

glutamine for 1 h at 37 °C in a non-CO2 incubator. Oxygen consumption and extracellular acidification were analyzed under basal conditions and through sequential injections of oligomycin (0.5 μM), FCCP (1 μM), and rotenone + antimycin A (0.5 μM each). Data for each well were normalized to cell number as determined by crystal violet staining.

## Mitochondrial mass and ROS measurements

Cells were stained with MitoTracker Green (Thermo Fischer Scientific) in serum-free medium for 30 min at 37 °C and imaged by confocal microscopy. mtDNA copy number was measured in BJ-Ras-ER cells by qPCR (primers are described in Supplementary Data 5).

To measure reactive oxygen species (ROS), cells were treated with the MitoSOX Red superoxide indicator (Thermo Fischer Scientific) at a concentration of 5 μM for 30 min at 37 °C. Cells were washed twice with PBS. Fluorescence was measured with a BD LSR Fortessa Analyzer.

## Ribosome profiling

Proliferating and Senescent BJ-Ras-ER cells were washed with ice-cold PBS (100 μg/ml Cycloheximide (CHX)). Cell pellets we lysed in RP-Lysis Buffer (20 mM Tris-HCL pH 7.5, 10 mM MgCl2, 100 mM KCl, 1 % Triton-X 100, 2 mM DTT, 100 μg/ml CHX, 1x EDTA-free cOmplete EDTA-free Protease Inhibitor Cocktail, Merck). Cell lysates were digested with 1U/μl RNase I for 45 min at room temperature. Monosomes were enriched in sucrose gradients using a Beckmann Colter SW41Ti rotor at 36,000 rpm for 2 h. RNA was isolated following a standard Trizol-Chloroform extraction and size-selected using a 10 % denaturing urea polyacrylamide gel.

RPF library construction in brief: RNA was dephosphorylated using 5U of T4 PNK (NEB). Subsequently, pre-adenylated UMI-linkers were ligated to the RNA 3'end using 100 U T4 RNA Ligase 2, truncated K227Q (NEB). The residual linker was eliminated with 25U 5' Deadenylase and 15 U RecJf (NEB) for 60 min at 30 °C. Ribosomal RNA was depleted using a biotinylated rRNA oligo pool (Supplementary Data 5) in 1x SSC buffer (3 M NaCl, 300 mM trisodium citrate, pH 7). rRNA was pulled down using MyOne Streptavidin C1 DynaBeads (Thermo Fischer Scientific). RNA was reverse transcribed using the SuperScript III First-Strand Synthesis System (Thermo Fischer Scientific). cDNA was size-selected using an 8% denaturing urea polyacrylamide gel. cDNA was circularized using the CircLigase II Kit from Lucigen. Samples were subjected to PCR to introduce Illumina indexes, followed by size selection on an 8 % polyacrylamide gel. DNA concentration was measured with the Qubit DNA HS kit (Thermo Fischer Scientific). The final RPF libraries were single-end sequenced with an Illumina Next-Seq2000 P2 system.

## Cytokine array

The cytokine array was performed with the Proteome Profiler Human XL Cytokine Array Kit (ARY022B, R&D Systems). Briefly, the conditioned medium was collected in a serum-free medium in proliferating and senescent BJ-Ras-ER for 24 hours. The conditioned medium was filtered and incubated on the array overnight at 4 °C. Membranes were washed three times with 1X Wash Buffer for 10 min each. The array was then incubated with the antibody cocktail for 1 h at room temperature. Membranes were washed three times before adding the provided chemiluminescent detection reagents. Pixel intensity was quantified with the Quick Spots Image Analysis Software.

## In vivo senescence surveillance assay

Male C57BL/6 J mice, 8 weeks, old, were purchased from Charles River or bred in-house at the DKFZ Center for Preclinical Research facility. The procedure involved the co-delivery of three vectors into the mice via hydrodynamic injection. The vectors used were the SB transposase vector and two SB transposon vectors expressing

*Nras*^G12D-IRES-GFP and a miR-30 shRNA (targeting Renilla, *eIf5a*, or *Smox*). At the indicated time points, mice were sacrificed, and samples of liver tissue were collected for further analysis. Liver tissue samples were fixed in optimal cutting temperature compound and used for subsequent histological analyses. Mice were housed in standard laboratory conditions. The dark/light cycle was maintained at 12 h of light and 12 h of darkness. The ambient temperature was controlled at $22 \pm 2\,°C$ and relative humidity was kept at 45–65%. All mice had ad libitum access to food and water and were housed in groups of 3-5 per cage. All mice were maintained in a pathogen-free facility and used according to the German Cancer Research Center and following permission by the controlling government office (Regierungspräsidium Karlsruhe) according to the German Animal Protection Law, and in compliance with the EU Directive on Animal Welfare, Directive 2010/63/EU.

### Histopathology
After overnight fixation in 10 % buffered formalin, representative specimens of the liver were routinely dehydrated, embedded in paraffin, and cut into 4 µm-thick sections. Tissue sections were routinely stained with hematoxylin and eosin according to standard protocols.

### Immunohistochemistry with anti-CD45
After heat-induced antigen retrieval at pH6, FFPE tissue sections were incubated overnight with the primary antibody and blocked with hydrogen peroxide (Peroxidase-blocking solution, Dako REAL, Agilent Technologies, Inc., Santa Clara, USA). An anti-rat secondary antibody (SS LINK) conjugated to Biotin was applied (Biogenex, Fremont, USA), followed by Streptavidin-Alkaline-Phosphatase Label (Biogenex, Fremont, USA) The signal was visualized using AEC Single Solution. (Zytomed Systems. Berlin, Germany) as a chromogen. Details are given in the table.

### Immunohistochemistry with anti-GFP
After heat-induced antigen retrieval at pH6, FFPE tissue sections were incubated overnight with the primary antibody. An anti-rabbit secondary antibody conjugated to AP was applied (POLYVIEW® PLUS AP, Enzo Life Sciences, USA). The signal was visualized using Permanent AP Red (Zytomed Systems. Berlin, Germany) as a chromogen. Details are given in the table.

### Reporting summary
Further information on research design is available in the Nature Portfolio Reporting Summary linked to this article.

## Data availability
The data supporting the findings of this study are available from the corresponding authors upon request. The whole proteome and nascent proteome data have been deposited to the Proteome Xchange Consortium via the PRIDE partner repository.

The global proteome data are available with the identifier PXD034251.

The nascent proteome data are available with the identifier PXD034334.

The raw metabolomics data is available in MetaboLights under the identifier MTBLS10690.

The sequencing data from this study have been submitted to NCBI BioProject (http://www.ncbi.nlm.nih.gov/bioproject) under BioProject number PRJNA982716.

Source data for the figures and Supplementary Figures. are provided as a Source Data file. Source data are provided in this paper.

## Code availability
The code for differential ribosome codon reading (diricore) is available at https://github.com/A-X-Smitt/B250_diricore.

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

## Acknowledgements

We thank Claudia Scholl and Niels Halama for their advice and critical discussions. We thank Jesús Gil, Ana Banito, and Almut Shulze for sharing the reagents. We thank Tanja Poth from the CMCP - Center for Model System and Comparative Pathology, Institute of Pathology, University Hospital Heidelberg for the technical processing of tissue samples. This work was funded in part by grants of the European Research Council "DualRP" (ERC StG No. 759579) and the German Research Foundation (DFG 504774163 and DFG 545215964) to F.L.-P. D.F.T. is funded by the German Research Foundation (DFG) under TS 293/3-1 and a European Research Council Starting Grant "CrispSCNAs" (Grant No. 948172). X.J. is funded by the DKFZ Clinician Scientist Program, which is supported by the Dieter Morszeck Foundation. A.K. is supported by a fellowship of the Helmholtz International Graduate School. G.P. is sup-ported by the German-Israeli Helmholtz International Research School in Cancer Biology. C.C.A.R. is supported by the DKFZ International Postdoc Program.

## Author contributions

X.J., A.H.B., and F.L.-P. conceived the project, designed all the experiments, and wrote the manuscript. Methodology and data acquisition: X.J., A.H.B., G.P., R.D.P., T.B., S.G., E.A.Z., C.R.B., C.C.A.R., A.K. and D.A.-H. Manuscript revision: W.P., D.F.T., D.T., J.K. and F.L.-P.

## Funding

## Competing interests

The authors declare no competing financial interests.
