## [Peer Review File · Nature Communications]

P53-dependent hypusination of eIF5A affects mitochondrial translation and senescence immune surveillanceREVIEWER COMMENTS

Reviewer #1 (Remarks to the Author):

This study convincingly shows that high rates of protein synthesis in oncogene senescent cells rely on hypusinated eIF5A and polyamine metabolism which is dependent on p53. In particular mitochondrial ribosomal proteins require eIF5A for protein synthesis in these cells. Interestingly proliferating cells do not have the same requirements.

While the experiments are well conducted, controlled and described, a few comments here:

1) Zamoyska et al showed in a PNAS paper in T cells (Tan et al) that GC7 did not have the same effect as knocking out the enzymes that hypusinate eIF5A. It would be crucial to show some of the major findings by CRISPR or siRNA of DHPS or DOHH.

2) page 9, line 312 the qPCR confirmed that SASP protein transcription is dependent on eIF5A hypusination via mitochondrial translation. How does this work? Please comment.

3) I find it quite difficult to follow how the different enzymes in the polyamine pathway fit with the measured metabolites. Why is SPD not found to be higher when SMOX and Sat1 are increased? Please draw cartoon what happens in senescent cells in comparison to proliferating cells in regard to these enzymes and their metabolites.

4) All experiments have been conducted in transformed tumour cell lines, and not on aged cells, and yet the discussion and introduction cites mostly papers on aging and eIF5A. There is some confusing literature on OIS versus aging induced senescence and their discrepancies. Do these pathways operate similarly in these two types of senescence? It would be good to include a paragraph discussing why DFMO or GC7 do not work/do work in tumour therapy. And also please include one experiment on old & senescent/non-proliferating primary cells to see if the same pathways operate here.

Reviewer #2 (Remarks to the Author):

In this study, Authors have put forth to seek to validate the effect of eIF5A hypusination on mitochondrial translation in senescent cell. The authors claimed that p53 plays a pivotal role in eIF5A hypusination and is required for synthesizing mitochondrial ribosomal proteins, including the proteins involved in immune surveillance. Overall, the manuscript raises some interesting ideas, but the conclusion is based on insufficient evidence and many missing linkages between each result. If this manuscript is to be published, it is suggested to revise the title and recommend a major revision with more robust research results.

Below are the list of key concerns;

1. The author claimed that senescent cells synthesize more proteins than proliferating cells, and that the results are the most important results of the entire study and serve as the basis for all other experimental results. However, it is well known that senescent cells exhibit global translation repression, with some specific mRNA translation increments, as the author mentioned throughout the manuscript. Therefore, additional experiments are suggested to confirm whether senescent cells truly synthesize more protein through translation enhancement. For example, ribosome profiling and metabolic labeling with S35-met followed by gel electrophoresis could be good choices.
2. This is a question related to the previous discussion. It appears that there is no quantitative change in eIF5A hypusination observed in either control or 4-OHT cells. What could be the mechanism by which the same level of eIF5A protein and hypusination specifically contributes to global translation only in senescent cells? The increase in global translation due to eIF5A hypusination cannot be solely attributed to increased translation of mitochondrial ribosomal proteins. For instance, in Fig 5F, mitochondrial protein synthesis increased by about 1.4-fold with 4-OHT, but total protein synthesis increased by more than 4-fold. Overall, there is insufficient evidence to support the claim that eIF5A

hypusination specifically enhances global translation in senescent cells.

3. In Fig 3 and 4, is there any way to see the level of Spermine? If the lack of quantitative change in spermidine is due to SMOX, then a quantitative change in spermidine should be observed. In addition, although increase in the amount of mRNA for SMOX and SAT1 was shown, it is important to show whether the amount of protein actually increased. If the SMOX protein increased, direct evidence is needed to determine whether it converts Spermine to Spermidine.

4. In Fig 5e, the authors showed that translation of mitochondrial ribosomal proteins was inhibited under GC7 treatment. It will be important to see if the global translation also be repressed under these condition.

5. It is suggested to have a clearer description between global translation and mitochondrial translation. Figure 5 was intended to show that the translation of mitochondrial ribosomal proteins is influenced by eIF5A. However, the analysis of elongation codons is seem to be cytoplasmic translation analysis, and if it was mitochondrial translation analysis, more detailed expression would have been necessary.

6. Although author claimed that there is correlation between mitochondrial translation, SASP and ROS, more direct evidence is required to establish the correlation between eIF5A hypusination-dependent mitochondrial translation and quantitative changes in cytokine mRNA.

Reviewer #3 (Remarks to the Author):

In this interesting paper, Jiang, Baig and colleagues examine how hypusination of eIF5A affects protein synthesis during cellular senescence and the authors nicely discover SMOX is a key enzyme, directly targeted by p53, to maintain the level of spermidine and eIF5A hypusination and hence protein synthesis during senescence. Although it is known that eIF5A hypusination is important for the translation of mitochondrial proteins in other models, the authors show eIF5A hypusination regulates the translation of mitochondrial ribosomal proteins and is thus needed for mitochondrial translation in senescent cells. Finally the authors discover the link between eIF5A hypusination and immune surveillance.

The manuscript is well written, easy to follow and the results are clearly presented. I am supportive of the work in general, but several concerns need to be addressed prior to publication.

Major point: eIF5A hypusination and translation of mitochondrial proteins and ribosomal proteins

I understand that the authors focus on cellular senescence but as it is known that eIF5A hypusination regulates the translation of mitochondrial proteins, this study would benefit from including additional data to strengthen their findings, i.e., the importance of translation of mitochondrial ribosomal proteins amongst other mitochondrial proteins.

From this point of view, the authors suggest that 'eIF5A plays a critical role in regulating the translation of mitochondrial ribosomal proteins in senescent cells' because 1) the component of the large and small subunits of the mitochondrial ribosomes were reduced and 2) mRNA levels of mitochondrial ribosome proteins did not change but mRNA association with polysomes was reduced in senescent cells exposed to GC7.

My concerns are:

a) changes in mRNA association with polysomes are modest. In my view, mRNAs of MRPL11 and MRPS30 are still associated with polysomes (I assume that MRPL11 mRNAs change just from tri-some to di-some). Examining other mRNA species of MRPL and MPRS by qRT-PCR or even 'translatome'

analysis by RNA-Seq or pSILAC with AHA labeling proteomics to check the translation of all MRPL and MRPS mRNAs would strengthen the data.

b) I wonder whether the abundance of mitochondrial ribosome decreases or not in senescent cells treated with/without GC7.

c) Fig. 5C shows that only 12-hour-treatment of GC7 in senescent cells reduces the abundance of mitochondrial ribosomal proteins drastically. However, mitochondrial ribosomal proteins are not such 'short-lived' proteins compared to others in the previous papers using other cell lines or models (PMID: 21593866, PMID: 21937730, PMID: 34715012 etc...). I suspect degradation of MRPS and MRPL rather than suppression of translation of MRPS and MRPL in this situation. The author should examine this possibility.

d) Related to the comments above, please explain the reason the authors chose the timepoint of 12h for GC7 treatment for analyses shown in Figure 5a-e. In Figure 3, 12h treatment of GC7 reduced the hypusination of eIF5A very slightly (there was significantly different though). I also assume a slight reduction of protein synthesis at this time point (data not shown in Fig. 3d).

e) The authors showed that 'the synthesis of all detected mitochondrially encoded proteins was significantly suppressed following GC7 treatment (Supplementary Fig. 5f).' However, 4 proteins (MT-CO2/ MT-CO3/ MT-ND6/ MT-CYB) are highlighted in the figure. The authors should highlight all proteins encoded in mtDNA.

f) In the previous reports (PMID: 36057633, PMID: 31130465), many mitochondrial proteins are reduced when hypusination of eIF5A is inhibited. I wonder whether the authors also saw the same reduction in mitochondrial proteins in their quantitative proteomic analysis shown in Fig. 5a.

g) Those reports (PMID: 36057633, PMID: 31130465) are important previous studies and therefore should first appear in the Introduction section rather than the Discussion section.

h) Although the authors use GC7 to inhibit hypusination, the authors sometimes mention 'inhibiting eIF5A'. I think the authors should describe correctly to avoid misunderstanding.

Minor points

- The authors should include and describe the data of spermidine in Fig. 3e. The word 'spermidine' just appears in the Figure legend (line 966).

- Please discuss further the importance of elevated protein synthesis as a general feature of senescent cells in the Discussion section.

- Related to the comments above and Fig. 1, what kind of proteins are upregulated in senescent cells? Mitochondrial proteins and mitochondrial ribosomal proteins are also increased?

- There are typos particularly in the Method section line that should be corrected.

- line 492 Antibodies section is not completed.

- line 961-962 *P < 0.001 by Student's t-test; ***P < 0.001 by Student's t-test. The first one should be *P < 0.01?

-Fig. 5b; Two 'Mitochondrial translation' with different value... what is the difference between two 'Mitochondrial translation'?

- Supplementary tables should be in separate excel files.

Reviewer #4 (Remarks to the Author):

Overall, this manuscript is extremely well written. It is a rare pleasure to review a paper that is elegant to read and from which I learned something new during the review process. My recommendation is to publish this manuscript with minor revision, specified below.

1. The first page of the results discusses protein synthesis rate in senescent cells. The discussion includes previous references to this phenomenon, but it is not mentioned in the introduction which made it relatively jarring that this was the first set of results. It is recommended to note the importance of protein synthesis rate with the corresponding literature in the introduction to prepare the reader for the data they will be considering.

2. Line 527—"----- tool"

3. Please outline the SP3 protocol once in the methods section. There are many versions of this technique and the original cited paper is nearly a decade old, with updated guidelines published by the same authors (Hughes et. al.). This will assist any group in replicating the procedure or data

4. iBAQ is a poorly performing technique for quantification, it is essentially label-free quantification with a fancier name. There are many better ways to collect and analyze proteomic mass spectrometry data, such as the implemented SILAC method, TMT/iTRAQ, or DIA. However, in context of this manuscript and wealth of data supplementing the iBAQ data, the MS data and analysis technique are sufficient to support the conclusions and claims. This specific comment is simply to inform the author's future research rather than as a flaw of the work presented.

Response to Reviewers' Comments

We thank the Reviewers for their critical reading of the manuscript and are pleased they found the study interesting. In response to the helpful suggestions, we now provide substantial new data strengthening our mechanistic insight into the function of eIF5A in regulating mitochondrial translation in senescent cells. Below are detailed responses for each comment.

Reviewer #1 (Remarks to the Author):

This study convincingly shows that high rates of protein synthesis in oncogene senescent cells rely on hypusinated eif5a and polyamine metabolism which is dependent on p53. In particular mitochondrial ribosomal proteins require eif5a for protein synthesis in these cells. Interestingly proliferating cells do not have the same requirements. While the experiments are well conducted, controlled and described, a few comments here:

1) Zamoyska et al showed in a PNAS paper in T cells (Tan et al) that GC7 did not have the same effect as knocking out the enzymes that hypusinate eif5a. It would be crucial to show some of the major findings by CRISPR or siRNA of DHPS or DOHH.

Response: We thank the reviewer for this comment. To address this point, we transduced sgRNAs targeting DHPS or DOHH into proliferating and senescent BJ-Ras-ER cells and assessed hypusination levels by western blot. Consistent with the findings from the GC7 treatment (Fig. 3b), we found that depletion of DHPS and DOHH results in reduced levels of eIF5A hypusination only in OIS cells.

Furthermore, we assessed protein synthesis rates in this system and found that the knockout of DHPS and DOHH leads to decreased rates of protein synthesis in senescent cells. These results have been incorporated into the revised manuscript and are presented in Supplementary Fig. 3a-b.

2) page 9, line 312 the qPCR confirmed that SASP protein transcription is dependent on eif5a hypusination via mitochondrial translation. How does this work? Please comment.

Response: Our data indicates that inhibition of eIF5A hypusination reduces the expression of SASP proteins in conditioned media from OIS BJ-Ras-ER cells (Fig. 6a-b). Furthermore, we assessed whether the expression of selected cytokine mRNAs was also decreased after GC7 treatment and established a link between inhibition of mitochondrial translation and SASP production.

Previous work showed that removing mitochondria in different models of cellular senescence attenuated their pro-inflammatory phenotype. This work suggested that the SASP is dependent on mitochondria (Correia-Melo C. et al. 2016; *EMBO J.*); however, the underlying mechanisms are not yet fully elucidated. Our study provides evidence that eIF5A activity is required for the synthesis of mitochondrial ribosomal proteins, affecting mitochondrial translation and therefore SASP production.

In the revised version of our manuscript, we modified the text of this section to clarify these points.

3) I find it quite difficult to follow how the different enzymes in the polyamine pathway fit with the measured metabolites. Why is spd not found to be higher when SMOX and Sat1 are increased? Please draw cartoon what happens in senescent cells in comparison to proliferating cells in regard to these enzymes and their metabolites.

Response: In the revised version of our manuscript, we depict the rewiring of the polyamine pathway in senescent cells in Supplementary Fig. 6f. Based on our findings, we illustrate that

in proliferating cells, the abundance of putrescine is sufficient to synthesize spermidine and maintain eIF5A hypusination levels. Conversely, in senescent cells, levels of putrescine significantly decrease via an unknown mechanism. The senescent program activates the expression of SMOX and SAT1 to recycle polyamines, thereby sustaining spermidine and eIF5A hypusination levels.

4) All experiments have been conducted in transformed tumour cell lines, and not on aged cells, and yet the discussion and introduction cites mostly papers on aging and eif5a. There is some confusing literature on OIS versus aging induced senescence and their discrepancies. Do these pathways operate similarly in these two types of senescence? It would be good to include a paragraph discussing why DFMO or GC7 do not work/do work in tumour therapy. And also please include one experiment on old & senescent/non-proliferating primary cells to see if the same pathways operate here.

Response: To investigate whether similar pathways are operational in the aging process, we utilized a cellular model of replicative senescence. In this model, prolonged replication of primary IMR-90 cells resulted in senescence after 49 population doublings (PD), as evidenced by the induction of p21 (Supplementary Fig. 3f). We observed an increase in the rates of protein synthesis in IMR-90 cells undergoing replicative senescence. Additionally, qPCR analysis revealed a significant upregulation of SAT1 and SMOX mRNAs under replicative senescence, suggesting that the activation of this pathway is a general characteristic of senescence. These new findings have been incorporated into Fig. 1f and Supplementary Fig. 3f.

Furthermore, in response to the suggestion, we discuss about the impact of polyamine synthesis inhibition in tumor therapy.

Reviewer #2 (Remarks to the Author):

In this study, Authors have put forth to seek to validate the effect of eIF5A hypusination on mitochondrial translation in senescent cell. The authors claimed that p53 plays a pivotal role in eIF5A hypusination and is required for synthesizing mitochondrial ribosomal proteins, including the proteins involved in immune surveillance. Overall, the manuscript raises some interesting ideas, but the conclusion is based on insufficient evidence and many missing linkages between each result. If this manuscript is to be published, it is suggested to revise the title and recommend a major revision with more robust research results.

Below are the list of key concerns;

1. The author claimed that senescent cells synthesize more proteins than proliferating cells, and that the results are the most important results of the entire study and serve as the basis for all other experimental results. However, it is well known that senescent cells exhibit global translation repression, with some specific mRNA translation increments, as the author mentioned throughout the manuscript. Therefore, additional experiments are suggested to confirm whether senescent cells truly synthesize more protein through translation enhancement. For example, ribosome profiling and metabolic labeling with S35-met followed by gel electrophoresis could be good choices.

Response: We appreciate the reviewer for this observation and concur that additional experiments will provide further support for our findings. To address this, we used a complementary method that measures the rate of global protein synthesis through L-azidohomoalanine (L-AHA)-based click chemistry (Lee Y. et al., 2021, *Star Protocols*). Using this approach, we observed that OIS in BJ-Ras-ER and IMR90-Ras-ER fibroblasts robustly increase protein synthesis compared to proliferating cells (Figure R1).

Figure R1. L-AHA incorporation measured by flow cytometry in proliferating and OIS BJ-Ras-ER and IMR90-Ras-ER cells. CHX, cycloheximide; MFI, Median fluorescence intensity. Data represent mean \pm SD (n = 3); ***P < 0.001 by Student's t-test.

Our results reveal that, at the single-cell level, senescent cells exhibit a higher rate of protein synthesis compared to proliferating cells. We have clarified this aspect in the revised version of the manuscript. Furthermore, consistent with our findings, recent evidence supports the notion that increased protein synthesis is a common phenomenon in senescent cells (Lee Y. et al., 2021, Dev. Cell; Roh, K. et al., Nat Metab., 2023; Dörr-JR et al., Nature, 2013).

2. This is a question related to the previous discussion. It appears that there is no quantitative change in eIF5A hypusination observed in either control or 4-OHT cells. What could be the mechanism by which the same level of eIF5A protein and hypusination specifically contributes to global translation only in senescent cells? The increase in global translation due to eIF5A hypusination cannot be solely attributed to increased translation of mitochondrial ribosomal proteins. For instance, in Fig 5F, mitochondrial protein synthesis increased by about 1.4-fold with 4-OHT, but total protein synthesis increased by more than 4-fold. Overall, there is insufficient evidence to support the claim that eIF5A hypusination specifically enhances global translation in senescent cells.

Response: We thank the reviewer for pointing this out. In general terms, we agree with the observations made:

1. While protein synthesis is increased in multiple cellular models of senescence, this cannot be solely attributed to eIF5A or the polyamine pathway. There may be a different unknown mechanism that enhances protein production during senescence. However, our CRISPR screen and validation data indicate that eIF5A is required to sustain the increased protein synthesis rates in senescent cells. We have amended our manuscript to ensure that all statements reflect this conclusion, and we elaborate on this issue in the discussion.

2. Regarding Fig 5F, we concur with the reviewer; there is a discrepancy between the fold change of mitochondrial protein synthesis and total protein synthesis. This is most likely due to the fact that total protein synthesis measurements quantify protein synthesis by cytosolic ribosomes and, to some extent, mitochondrial ribosomes, while the experiment in Fig. 5F measures protein synthesis exclusively in the mitochondrial compartment. We have modified our manuscript to clarify this point.

3. In Fig 3 and 4, is there any way to see the level of Spermine? If the lack of quantitative change in spermidine is due to SMOX, then a quantitative change in spermidine should be observed. In addition, although increase in the amount of mRNA for SMOX and SAT1 was shown, it is important to show whether the amount of protein actually increased. If the SMOX protein increased, direct evidence is needed to determine whether it converts Spermine to Spermidine.

Response: To demonstrate the increased levels of the SMOX protein during OIS, we conducted western blot analysis. Our results reveal higher expression of SMOX protein in

senescent cells. Transfection of SMOX siRNAs in senescent cells resulted in the downregulation of protein expression.

Due to technical limitations, the detection of spermine and acetyl-spermine through our LC/MS-based metabolomics analysis proved challenging. To overcome this, we used a Spermine ELISA Kit, allowing for the in vitro quantification of spermine concentrations. Using this alternative approach, we observed a reduction in spermine levels in OIS BJ-Ras-ER cells. Interestingly, SMOX knockdown in senescent cells restored intracellular spermine levels. Taken together, our results support the conclusion that p53-mediated transcriptional activation of SMOX contributes to sustaining spermidine levels and eIF5A hypusination during senescence. These findings have been incorporated into Supplementary Fig. 4b-c.

4. In Fig 5e, the authors showed that translation of mitochondrial ribosomal proteins was inhibited under GC7 treatment. It will be important to see if the global translation also be repressed under these condition.

Response: To assess whether global translation is repressed under these conditions, we conducted an OP-Puro assay on OIS BJ-Ras-ER cells treated with GC7. We observed a time-dependent decrease in global protein synthesis in senescent fibroblasts. These data have been incorporated into Fig. 3d of the revised manuscript.

5. It is suggested to have a clearer description between global translation and mitochondrial translation. Figure 5 was intended to show that the translation of mitochondrial ribosomal proteins is influenced by eIF5A. However, the analysis of elongation codons is seem to be cytoplasmic translation analysis, and if it was mitochondrial translation analysis, more detailed expression would have been necessary.

Response: We thank the reviewer for this suggestion. The ribosome profiling data in Fig. 5k-l and Supplementary Fig. 5k-l specifically depict cytoplasmic translation. We have clarified this aspect in both the main text and the figure legends of our revised paper.

6. Although author claimed that there is correlation between mitochondrial translation, SASP and ROS, more direct evidence is required to establish the correlation between eIF5A hypusination-dependent mitochondrial translation and quantitative changes in cytokine mRNA.

We thank the reviewer for providing this comment. As stated in our manuscript, dysfunctional mitochondrial translation leads to an increase in the production of reactive oxygen species (ROS), thereby adversely affecting the SASP (Correia-Melo et al., 2016, EMBO J; Martini & Passos, 2022, FEBS J.; Wiley et al., 2016, Cell Metab).

To establish a strong correlation between eIF5A hypusination, mitochondrial dysfunction, and ROS levels, we conducted measurements of mitochondrial ROS in proliferating and OIS BJ-Ras-ER cells treated with either DMSO or GC7. Our findings reveal that senescent cells exhibit elevated mitochondrial ROS levels compared to proliferating cells, and this effect is exacerbated by the inhibition of eIF5A hypusination. Conversely, GC7 treatment has no impact on mitochondrial ROS production in proliferating cells. These results have been integrated into Supplementary Fig 6a of our revised manuscript.

Furthermore, inhibition of eIF5A hypusination, SMOX inhibition, and inhibition of mitochondrial translation lead to a robust reduction in the expression of SASP factors (Fig. 6c and Supplementary Fig. 6c-e). Taken together, these findings indicate that inhibition of eIF5A hypusination results in dysfunctional mitochondrial translation, increasing ROS levels and adversely affecting the SASP.

Reviewer #3 (Remarks to the Author):

In this interesting paper, Jiang, Baig and colleagues examine how hypusination of eIF5A affects protein synthesis during cellular senescence and the authors nicely discover SMOX is a key enzyme, directly targeted by p53, to maintain the level of spermidine and eIF5A hypusination and hence protein synthesis during senescence. Although it is known that eIF5A hypusination is important for the translation of mitochondrial proteins in other models, the authors show eIF5A hypusination regulates the translation of mitochondrial ribosomal proteins and is thus needed for mitochondrial translation in senescent cells. Finally the authors discover the link between eIF5A hypusination and immune surveillance.

The manuscript is well written, easy to follow and the results are clearly presented. I am supportive of the work in general, but several concerns need to be addressed prior to publication.

Major point: eIF5A hypusination and translation of mitochondrial proteins and ribosomal proteins

I understand that the authors focus on cellular senescence but as it is known that eIF5A hypusination regulates the translation of mitochondrial proteins, this study would benefit from including additional data to strengthen their findings, i.e., the importance of translation of mitochondrial ribosomal proteins amongst other mitochondrial proteins.

From this point of view, the authors suggest that 'eIF5A plays a critical role in regulating the translation of mitochondrial ribosomal proteins in senescent cells' because 1) the component of the large and small subunits of the mitochondrial ribosomes were reduced and 2) mRNA levels of mitochondrial ribosome proteins did not change but mRNA association with polysomes was reduced in senescent cells exposed to GC7.

My concerns are:

a) changes in mRNA association with polysomes are modest. In my view, mRNAs of MRPL11 and MRPS30 are still associated with polysomes (I assume that MRPL11 mRNAs change just from tri-some to di-some). Examining other mRNA species of MRPL and MRPS by qRT-PCR or even 'translatome' analysis by RNA-Seq or pSILAC with AHA labeling proteomics to check the translation of all MRPL and MRPS mRNAs would strengthen the data.

Response: We thank the reviewer for bringing this to our attention. The observed moderate changes in mRNA association with polysomes in senescent cells may be attributed to the duration of the GC7 treatment, specifically 12 hours in this experiment. To explore this, we subjected OIS BJ-Ras-ER cells to a 24-hour treatment and conducted polysome analysis for the same targets. Under these conditions, a more pronounced displacement of mitochondrial ribosomal protein mRNAs towards the non-polysomal fractions was noted, whereas mRNA association in proliferating cells remained unaltered. These findings have been incorporated into Fig. 5e and Supplementary Fig. 5d of the revised manuscript.

Additionally, another MRPL mRNA, MRPL37, exhibited a similar response (Figure R2). Altogether, these data support the conclusion that the translation of mitochondrial ribosomal proteins in senescent cells is dependent of eIF5A activity.

Figure R2. Polysome association of MRPL37 mRNA in OIS and proliferating BJ-Ras-ER cells treated with GC7 (10 μ M) for 24 hours. The data shows mean \pm SD (n = 3). NS, not significant; ***P < 0.001 by Student's t-test.

b) I wonder whether the abundance of mitochondrial ribosome decreases or not in senescent cells treated with/without GC7.

Response: To evaluate whether the abundance of translating mitochondrial ribosomes decreases in senescent cells treated with GC7, we assessed the levels of 16S mitochondrial rRNA in sucrose gradients. The association of 16S rRNA with polysomal fractions serves as a measure of the number of translating mito-ribosomes. Treatment with GC7 significantly altered the distribution of 16S rRNA, shifting it towards the free RNA fractions, while the total levels remained unaffected by the treatment (Figure R2). Collectively, these results, in conjunction with the whole proteomics data and western blot validations, indicate that the inhibition of eIF5A activity in senescent cells leads to a decrease in the translation of mitochondrial ribosomal proteins and the abundance of translating mito-ribosomes.

Figure R3. qRT-PCR quantification of mitochondrial 16S rRNA in OIS BJ-Ras-ER cells treated with GC7 (10 μ M) for 24 hours, both in sucrose gradients (left panel) and total lysates (right panel). SS denotes the small subunit of the mito-ribosome, while LS refers to the large subunit of the mitoribosome.

c) Fig. 5C shows that only 12-hour-treatment of GC7 in senescent cells reduces the abundance of mitochondrial ribosomal proteins drastically. However, mitochondrial ribosomal proteins are not such 'short-lived' proteins compared to others in the previous papers using other cell lines or models (PMID: 21593866, PMID: 21937730, PMID: 34715012 etc...). I suspect degradation of MRPS and MRPL rather than suppression of translation of MRPS and MRPL in this situation. The author should examine this possibility.

Response: To address this point, we conducted a cycloheximide (CHX) chase assay in OIS BJ-Ras-ER cells treated with a control or GC7 for 24 hours. We evaluated the protein levels of

mitochondrial ribosomal proteins MRPL37 and MRPS33. As depicted in Fig. 5c, we noted a reduction in the expression of both proteins following the inhibition of hypusination. However, protein stability remained unchanged in all conditions. These findings suggest that the alterations in protein levels observed after GC7 treatment are primarily attributed to reduced translation of mitochondrial ribosomal proteins. These data were incorporated in Fig. 5f of our revised manuscript.

d) Related to the comments above, please explain the reason the authors chose the timepoint of 12h for GC7 treatment for analyses shown in Figure 5a-e. In Figure 3, 12h treatment of GC7 reduced the hypusination of eIF5A very slightly (there was significantly different though). I also assume a slight reduction of protein synthesis at this time point (data not shown in Fig. 3d).

Response: The rationale behind choosing a 12-hour time point in the proteomics analysis (Fig. 5a) is that this represents the earliest instance where changes in hypusination and global rates of protein synthesis become apparent. Our objective was to identify proteins most susceptible to the inhibition of hypusination exclusively in senescent cells. However, it is noteworthy that the majority of validation assays were conducted 24 hours after treatment (Fig. 5c-g). To validate that protein synthesis diminishes after a 12-hour GC7 treatment in senescent cells, we have incorporated this time point into the experiment presented in Fig. 3d.

e) The authors showed that ‘the synthesis of all detected mitochondrially encoded proteins was significantly suppressed following GC7 treatment (Supplementary Fig. 5f).’ However, 4 proteins (MT-CO2/ MT-CO3/ MT-ND6/ MT-CYB) are highlighted in the figure. The authors should highlight all proteins encoded in mtDNA.

Response: We appreciate the reviewer for bringing this to our attention. Our proteomics analysis faced limitations in detecting all proteins encoded in the mitochondrial genome, possibly due to the efficiency of AHA incorporation in mitochondria. Despite this challenge, our method consistently identified four proteins out of the thirteen encoded in mtDNA, all of which exhibited decreased translation following GC7 treatment.

f) In the previous reports (PMID: 36057633, PMID: 31130465), many mitochondrial proteins are reduced when hypusination of eIF5A is inhibited. I wonder whether the authors also saw the same reduction in mitochondrial proteins in their quantitative proteomic analysis shown in Fig. 5a.

Response: In Figure R4, we compare the expression changes of the mitochondrial proteins described in both studies between proliferating (Control) and OIS cells (4-OHT) treated with GC7. Our analysis shows no significant changes in the levels of these proteins following GC7 treatment. Notably, the studies by Zhou J. et al. and Puleston D.J. et al. utilized mouse liver tissue and mouse macrophages, respectively, suggesting that the observed effects might be cell-type specific. We discuss on this point in the revised version of our manuscript.

Figure R4. Differential protein expression after GC7 (10 μ M) treatment in proliferating BJ-Ras-ER cells (Control) and OIS- BJ-Ras-ER cells (4-OHT).

g) Those reports (PMID: 36057633, PMID: 31130465) are important previous studies and therefore should first appear in the Introduction section rather than the Discussion section.

Response: This is an important point. We have cited these references in the Introduction section accordingly.

h) Although the authors use GC7 to inhibit hypusination, the authors sometimes mention 'inhibiting eIF5A'. I think the authors should describe correctly to avoid misunderstanding.

Response: We thank the reviewer for bringing this to our attention. We have rectified the terminology throughout the manuscript.

Minor points

- The authors should include and describe the data of spermidine in Fig. 3e. The word 'spermidine' just appears in the Figure legend (line 966).

Response: In the revised version of the paper, we describe the spermidine data in page 6, line 184

- Please discuss further the importance of elevated protein synthesis as a general feature of senescent cells in the Discussion section.

Response: We have included a new paragraph discussing this point.

- Related to the comments above and Fig. 1, what kind of proteins are upregulated in senescent cells? Mitochondrial proteins and mitochondrial ribosomal proteins are also increased?

Response: The majority of proteins upregulated in senescent cells are associated with the Reactome terms such as cellular senescence, the hallmark of G2M checkpoint, and nucleosome organization (Figure R5). Additionally, we noted a modest, but not significant, increase in the expression of mitochondrial ribosomal proteins. This analysis implies that the predominant proteins upregulated in OIS cells are linked to the senescent program. However, the group most sensitive to hypusination inhibition is related to mitochondrial translation. We elaborate on this point in the discussion section of the revised version of our paper.

Figure R5. Volcano plot of the global proteome of proliferating and senescent BJ fibroblasts. Proteins were quantified with ≥ 2 LFQ intensities in 3 replicates of each condition. Protein groups with a log2 fold change > 1.0 and adj. p-value < 0.01 , representing proteins with higher abundance in senescent cells are highlighted in red. Proteins with a log2 fold change < -1.0 and adj. p-value < 0.01 , representing proteins with reduced abundance in senescent cells are highlighted in blue.

- There are typos particularly in the Method section line that should be corrected.

Response: In the latest version of our manuscript, we have corrected the typos of this section

- line 492 Antibodies section is not completed.

Response: We thank the reviewer for pointing this out. We have completed the missing information in the revised version.

*- line 961-962 *P < 0.001 by Student's t-test; ***P < 0.001 by Student's t-test. The first one should be *P < 0.01?*

Response: The reviewer is correct; we have rectified this in the revised version.

-Fig. 5b; Two 'Mitochondrial translation' with different value... what is the difference between two 'Mitochondrial translation'?

Response: The two mitochondrial translation pathways represent Reactome (1) and Gene Ontology (2) terms: R-HAS-5368287 (94 proteins) and GO:0032543 (109 proteins), respectively. This information has been incorporated into the legend of Fig. 5b.

- Supplementary tables should be in separate excel files.

Response: The revised version of our manuscript includes the supplementary tables as separate Excel files.

Reviewer #4 (Remarks to the Author):

Overall, this manuscript is extremely well written. It is a rare pleasure to review a paper that is elegant to read and from which I learned something new during the review process. My recommendation is to publish this manuscript with minor revision, specified below.

1. The first page of the results discusses protein synthesis rate in senescent cells. The discussion includes previous references to this phenomenon, but it is not mentioned in the introduction which made it relatively jarring that this was the first set of results. It is recommended to note the importance of protein synthesis rate with the corresponding literature in the introduction to prepare the reader for the data they will be considering.

Response: We thank the reviewer for the recommendation. In the revised version of our manuscript, we elaborate on the importance of protein synthesis in senescent cells in the introductory section.

2. Line 527—"----- tool"

Response: We apologize for the typo. The section has been revised, and we now include information about the shRNA designing tool.

3. Please outline the SP3 protocol once in the methods section. There are many versions of this technique and the original cited paper is nearly a decade old, with updated guidelines published by the same authors (Hughes et. al.). This will assist any group in replicating the procedure or data

Response: We agree with the reviewer regarding the importance of detailing the SP3 protocol. In the revised version of the manuscript, we provide a comprehensive description of the protocol.

4. iBAQ is a poorly performing technique for quantification, it is essentially label-free quantification with a fancier name. There are many better ways to collect and analyze proteomic mass spectrometry data, such as the implemented SILAC method, TMT/iTRAQ, or DIA. However, in context of this manuscript and wealth of data supplementing the iBAQ data, the MS data and analysis technique are sufficient to support the conclusions and claims. This specific comment is simply to inform the author's future research rather than as a flaw of the work presented.

Response: We appreciate the reviewer for bringing this to our attention, and we welcome the suggestion to explore an alternative method for analyzing the proteomic data.

REVIEWER COMMENTS

Reviewer #2 (Remarks to the Author):

The author has addressed many of the reviewers' concerns and questions, and there have been significant improvements to the manuscript. However, the paper's title and final conclusions still do not appear clear.

Additionally, this regulation is described as affecting immune surveillance, but this appears to be an overinterpretation of the results. Upon reviewing the entire paper, it is clear that the most significant effect observed when eIF5A is knocked down (KD) or knocked out (KO) in senescent cells, or when treated with GC7 to reduce Hyp-eIF5A expression, is on global translation, and this likely also affects the various cytokines seen in figure 6.

A proteomic analysis under 4-OHT conditions showed a significant increase in PI3K/AKT, and indeed, global translation increased under these conditions. Furthermore, treating with GC7 to inhibit eIF5A hypusination resulted in reduced global translation, suggesting that eIF5A regulates both global and mitochondrial translation.

Moreover, there is a lack of direct evidence to explain the correlation between mitochondrial translation regulation and immune surveillance. The mere change in mitochondrial translation and the variation in various cytokine expressions do not establish a direct link, and since global translation is also affected, the paper's title and conclusions seem to be an overinterpretation.

Additionally, the author needs to clearly explain how mitochondrial specific translation was observed separately from global translation.

Reviewer #3 (Remarks to the Author):

The authors have carefully addressed my concerns and conducted additional experiments. I have no further comments for authors. Thank you.

Reviewer #4 (Remarks to the Author):

the authors addressed my concerns.

Response to Reviewers' Comments

We are grateful for the reviewers' comments and suggestions, which have helped us to improve our work. In response to their helpful suggestions, we have modified the title and main text of the manuscript. Below are detailed responses to their comments:

Reviewer #2 (Remarks to the Author):

The author has addressed many of the reviewers' concerns and questions, and there have been significant improvements to the manuscript. However, the paper's title and final conclusions still do not appear clear.

1) Additionally, this regulation is described as affecting immune surveillance, but this appears to be an overinterpretation of the results. Upon reviewing the entire paper, it is clear that the most significant effect observed when eIF5A is knocked down (KD) or knocked out (KO) in senescent cells, or when treated with GC7 to reduce Hyp-eIF5A expression, is on global translation, and this likely also affects the various cytokines seen in figure 6.

A proteomic analysis under 4-OHT conditions showed a significant increase in PI3K/AKT, and indeed, global translation increased under these conditions. Furthermore, treating with GC7 to inhibit eIF5A hypusination resulted in reduced global translation, suggesting that eIF5A regulates both global and mitochondrial translation.

Moreover, there is a lack of direct evidence to explain the correlation between mitochondrial translation regulation and immune surveillance. The mere change in mitochondrial translation and the variation in various cytokine expressions do not establish a direct link, and since global translation is also affected, the paper's title and conclusions seem to be an overinterpretation.

Response: We understand the concern regarding the title and conclusion. Accordingly, we have revised the title to "P53 dependent hypusination of eIF5A affects mitochondrial translation and senescence immune surveillance." Additionally, we have modified the relevant sentence in the abstract to: "Our findings establish an important role of protein synthesis during cellular senescence and suggest a link between eIF5A, polyamine metabolism, and senescence immune surveillance."

2) Additionally, the author needs to clearly explain how mitochondrial specific translation was observed separately from global translation.

Response: We thank the reviewer for pointing this out. In the main text of the revised version of the manuscript, we have added a detailed explanation of how mitochondrial-specific translation was observed separately from global translation. Specifically, we include the following description (lines 272-279): "We measured global rates of mitochondrial translation separately from cytosolic translation. For this purpose, we treated cells with OP-Puro, which is incorporated into mitochondrial nascent polypeptide chains, along with MitoTracker Deep Red. We then isolated mitochondria, conjugated OP-Puro to a fluorochrome via Click chemistry, and measured both signals using flow cytometry."